# Microstructural Stability of the CoCrFe₂Ni₂ High Entropy Alloys with Additions of Cu and Mo

**Isaac Toda-Caraballo** [1,*] , **Jose Antonio Jiménez** [1] , **Srdjan Milenkovic** [2] , **Jorge Jimenez-Aguirre** [1,3] **and David San-Martín** [1]

1  Materalia Group, National Center for Metallurgical Research (CENIM-CSIC), Avda. Gregorio del Amo 8, 28040 Madrid, Spain; jimenez@cenim.csic.es (J.A.J.); jorge.jimeneza@upm.es (J.J.-A.); dsm@cenim.csic.es (D.S.-M.)
2  IMDEA Materials Institute, Tecnogetafe, Calle Eric Kandel 2, 28906 Getafe, Spain; srdjan.milenkovic@imdea.org
3  CIME (Centro de Investigación en Materiales Estructurales), Materials Science Department, Universidad Politécnica de Madrid, E.T.S. Ingenieros de Caminos, 28040 Madrid, Spain
*  Correspondence: isaac.toda@cenim.csic.es; Tel.: +34-915-53-89-00 (ext. 445154)

**Abstract:** New High Entropy Alloys based on the CoCrFe₂Ni₂ system have been developed by adding up to 10 at. % of Cu, Mo, and Cu + Mo in different amounts. These alloys showed a single face-centred cubic (FCC) structure after homogenization at 1200 °C. In order to evaluate their thermal stability, aging heat treatments at 500, 700, and 900 °C for 8 h were applied to study the possible precipitation phenomena. In the alloys where only Cu or Mo was added, we found the precipitation of an FCC Cu-rich phase or the μ phase rich in Mo, respectively, in agreement with some of the results previously shown in the literature. Nevertheless, we have observed that when both elements are present, Cu precipitation does not occur, and the formation of the Mo-rich phase is inhibited (or delayed). This is a surprising result as Cu and Mo have a positive enthalpy of mixing, being immiscible in a binary system, while added together they improve the stability of this system and maintain a single FCC crystal structure from medium to high temperatures

**Keywords:** high entropy alloys; alloy design; thermal treatments; microstructural characterization

## 1. Introduction

The CoCrFeNi system has been extensively studied in the HEA (High Entropy Alloy) scientific community, and it is the base composition of a large number of multicomponent alloys or compositionally complex multi-principal alloys, depending on the nomenclature employed. The excellent combination of these four elements in terms of the atomic size, enthalpy of mixing, and electronegativity makes them of paramount importance in this field. The CoCrFeNi system has a good microstructural stability [1–3], although the mechanical properties obtained are in general very limited [2,4,5]. In consequence, many works have explored different elemental additions, either with the intention of enhancing its mechanical response or improving other properties. As a result, more than 300 alloys have been analyzed and reported in the literature on this system [6], obtaining a variety of strengthening effects. Some examples are solid-solution hardening [7], precipitation hardening [8], or even exploring the TRansformation Induced Plasticity (TRIP) and/or the TWinning Induced Plasticity (TWIP) effects [9], as well as many other properties, such as oxidation [10], corrosion [11], improvements in irradiation resistance [12] (without excessive compromise of its microstructural stability), or recent studies concerning their processability [13,14].

Among the most common additions, Mn is of paramount importance and forms the very well-known Cantor alloy CoCrFeMnNi [15], extensively explored in the literature (see the review at [6]). Of special relevance in this system is the observation of deformation

twinning [9], demonstrating that HEAs could benefit from this hardening effect. A compositional modification of the Cantor alloy, which followed a reduction in the stacking fault energy in the system, showed that this effect can be controlled [16,17], paving the way to explore further compositions to tailor this advantageous effect.

Al is another element of great relevance and forms a separate family of HEAs. Al not only induces a strong hardening effect, but also a microstructural change in the system as Al is a strong body-centred cubic (BCC) stabilizer. The CoCrFeNi system displays a face-centred cubic (FCC) crystal structure, as long as the four elements are nearly equiatomic, which is transformed into FCC + BCC and finally into a BCC or B2 phase for x ≥ 1 in the $Al_xCoCrFeNi$ [18] system. This provides a large freedom for tailoring the microstructure, and hence its mechanical properties, as a full gradient of the volume fractions of FCC and BCC phases can be designed. It is not surprising then that this system is among the most studied in the literature.

Similarly populating the literature is the use of Cu in the CoCrFeNi system. The principal use of Cu is to stabilize the FCC structure. Cu nevertheless easily segregates, forming Cu-rich precipitates in this system [11,19–21] and others, such as in the AlCoCrFeNi + Cu [22–24], the AlCoCrFeNiTi + Cu [25], or the AlCoNiTiZn + Cu [26] families. This highlights the inclination of this element to elemental separation, which occurs especially at the grain boundaries, often inducing a detrimental effect into the mechanical properties or even into the microstructural stability due to the matrix compositional variations. Al is also commonly added to the CoCrCuFeNi system, providing single FCC [27,28] or single BCC solid solutions [29], depending on the Al addition, but also FCC + BCC dual-phase HEAs [30,31]. The transition between FCC and BCC occurs for x ≥ 2.5 in the $Al_xCoCrCuFeNi$ system [32], in contrast with x ≥ 1 in the $Al_xCoCrFeNi$ system, as cited above. This highlights the strong FCC stabilizing effect of Cu in the CoCrFeNi system. Additionally, other phases, such as B2, $L1_2$, or σ phases [33,34], may appear, increasing the complexity of the AlCoCrCuFeNi system along with its composition. Other common additions to the CoCrCuFeNi system are Ti [21,25], showing two FCC phases (one Cu-rich phase usually) along with the precipitation of the Laves phases but also Al + V [35], which may form FCC + BCC dual-phase structures and the potential presence of a σ phase.

The above-mentioned Cu-precipitation or segregation at the grain boundaries is an important issue in Cu-containing HEAs, especially in high-temperature applications, and it is always attributed to a positive enthalpy with respect to the other elements [36]. The thermodynamics of the alloy may vary significantly from the original solid-solution alloy [37], leading for instance to localized corrosion [11]. For as-cast microstructures with the addition of Mn, an increase in the yield strength is attributed to nano-size dispersed Mn-Cu particles [37], but at high temperatures, the mechanical properties have been seen to vary after aging Cu-containing HEAs, where, for instance, a decrease in hardness is associated at treatment temperatures between 500 and 900 °C [20]. In that work, the hardness increases again at treating temperatures above 1000 °C, which is attributed to the solid solutioning of Cu due to the larger entropic effect at higher temperatures. Moreover, a detrimental effect on wear resistance is found when the Cu content is increased in CoCrFeNi, which follows a complete separation of Cu in the microstructure [38].

On the other hand, the CoCrCuFeNi system is a very promising system to analyze due to its large microstructural stability, with FCC phase structures remaining unchanged in temperature ranges between 350 °C and 1350 °C [20], and interesting mechanical properties [27,39]. The processing simplicity and the elemental availability support its exploration in order to find alloys of industrial interest. Nevertheless, its potential application at high temperatures demands the preventing of the formation of precipitates and the Cu-rich FCC secondary phases cited above, as this may deeply influence the microstructure and/or even induce phase transitions. In this work, we explore different additions of Cu and Mo to a $CoCrFe_2Ni_2$ base alloy to study their influence on the microstructure and thermal stability. Interestingly, we found that small amounts of Mo impede the formation of Cu-rich

particles. On the other hand, we observed that Mo precipitation occurs for the Cu-free alloys, which, similarly, are impeded in the presence of Cu.

## 2. Alloy Design

In order to explore new elemental additions with respect to those already reported in the literature, there are elements such as Mo, Nb, V, and Ta that are easily available and good candidates because the enthalpies of formation of the Cu binary compounds are $\Delta H_{form}$ = 83, −29, 13, and 28 meV/atom, respectively [40], which are values reasonably close to 0, and hence may avoid the formation of Cu intermetallics. Other elements typically used in HEAs, such as Al or Ti, can be also considered; nevertheless, their corresponding enthalpies of formation with Cu are $\Delta H_{form}$ = −224 and −147 meV/atom, respectively. Additionally, such elements may also form other intermetallics, as it happens in the CoCrFeNi + Al and CoCrFeNi + Ti systems [21,25,33,34] due to the large binary compound negative enthalpy of Al and Ti with Co, Cr, and Fe, but especially with Ni ($\Delta H_{form}^{AlNi}$ = −428 meV/atom, $\Delta H_{form}^{TiNi}$ = −435 meV/atom [40]). On the other hand, large positive values may induce separation between the added element and Cu, which at the same time will not avoid Cu segregation. This was described in [41,42], where the reduction in the enthalpy of the mixing in combination with the strain energy is suggested to be the driving force for phase separation. This selecting criterion also explains the good microstructural behavior of CoCrFeNi systems after adding V and Mn, with $\Delta H_{form}$ = 13 and 29 meV/atom, respectively [40], as experimentally observed in [15,35].

A fast screening tool to help in the identification of potential segregation in the HEAs was proposed in [43], where the parameter λ accounts for the deviation of a certain composition with respect to the composition in such a system with the lowest solid-solution Gibbs free energy. It is worth noting that the parameter λ was experimentally assessed for the case of Cu segregation around the CoCrFeNi system. The higher the value of λ, the larger the probability of obtaining elemental segregation. In this work, and for the CrCoFeNi system, this parameter has been calculated for the cases where Cu-Mo, Cu-Nb, Cu-V, and Cu-Ta are added. Moreover, Cu-Al and Cu-Ti are considered, as they have been cited above.

The results are depicted in Figure 1 for the cases where Cu + X = 10 at. %, with X = Mo, Nb, V, Ta, Al, and Ti as possible candidates to be added as elemental additions. Even though this parameter serves only as an indicator, it is evident that the addition of Cu-Mo shows the lowest λ values among any other binary combination. Therefore, it is expected that the segregation in the CrCoFeNi + Cu + Mo may be lower than in any other system considered. A good candidate could be also V, although it is certainly much above Mo. Mo also shows the best fit with Co, Cr, Fe and Ni in terms of the enthalpy of mixing, and it displays good atomic size mismatch, electronegativity mismatch, and reasonable interatomic spacing mismatch [29,44–46]. This makes Mo a good choice to fulfil most of the solid-solution formation rules proposed in the literature. In addition to this, the large atomic radius of Mo will probably induce a solid-solution hardening effect from which the alloys will benefit [7].

Based on these criteria, the CoCrCuFeMoNi system will be explored in this work. To the best of the knowledge of the authors, this work represents the first investigation on this system. The research strategy first screens the base alloy, and it is followed by additions of Cu, Mo, and Cu + Mo in different contents, provided below. The microstructures of these alloys are compared after the homogenization and also after the application of thermal treatments at lower temperatures in order to determine the compositional window for obtaining stable single-solid solutions. To determine the base alloy, additional thermodynamic calculations were performed employing Thermo-Calc software and the TCHEA4 database (Thermo-Calc Software AB, Solna, Sweden with the High Entropy Alloy database version 4). In the calculations, the Cr content is required to be higher than 15 at. % to guarantee that oxidation protection is obtained, but low enough to minimize the σ-phase formation. The results at the allowed variations of Fe, Ni, and Co show that the predicted

segregation of Cu (secondary Cu-rich FCC phase) behaves quite similarly, irrespective of the base alloy content. A Cu content of 10 at. % is predicted to induce the presence of a Cu-rich FCC structure at temperatures around 1000–1100 °C, while the temperature decreases down to 600–700 °C for lower Cu contents (around 3 at. %). This occurs in all the base compositions, except for the cases with a high content of Co, where a substantial increase in the formation temperature of the Cu segregation occurs, even reaching the solidus temperature. The segregation is predicted in temperatures ranging approximately between 750 °C and 1100 °C. On the other hand, for compositions with a very large content of Fe, a σ-phase may form in large amounts, whereas for compositions with a large content of Ni, a μ-phase is also formed in large volumes. Therefore, a balance was found with the base composition $Co_{15}Cr_{15}Fe_{30}Ni_{30}$, in which the thermodynamic computations predict a single FCC solid solution from 575 °C until the melting temperature.

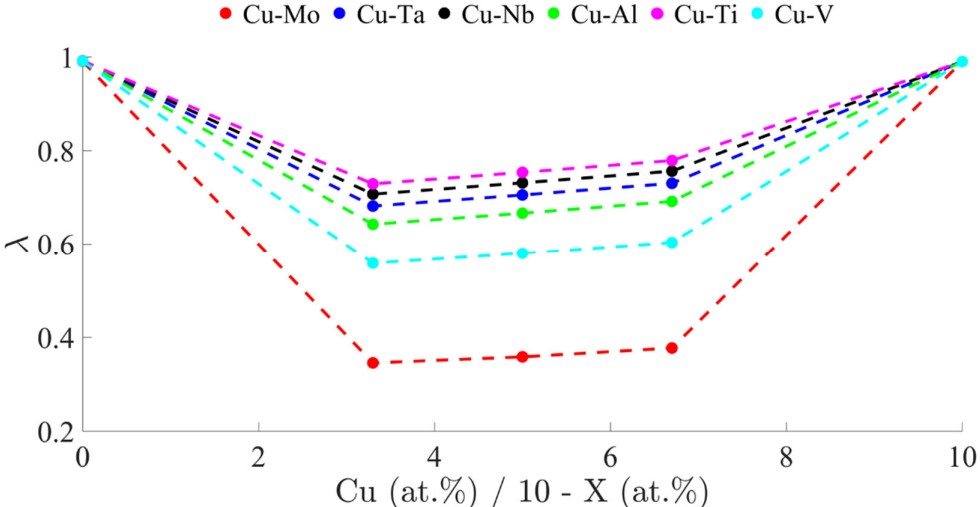

**Figure 1.** Calculated parameter λ as a function of Cu content and other elemental additions in the CrCoFeNi, where Cu + X = 10, and X corresponds to Mo, Ta, Nb, Al, Ti, and V.

This composition shows good behavior with regard to the principal solid-solution formation rules with atomic size mismatch $\delta$ = 1.16% and enthalpy of mixing $\Delta H_{mix}$ = −3.3 kJ/mol [29], interatomic spacing mismatch $s_m$ = 0.2785% [45], and electronegativity mismatch $\Delta \chi$ = 4.13% [46], while it displays a good fit with different combinations of Cu and Mo. Such combinations have been designed to cover a representative range of the Cu, Mo and Cu + Mo additions so that the influence of these elements can be properly analyzed. Additions of 10 at. % of Cu, $Cu_2Mo$, CuMo, $CuMo_2$, and Mo are considered.

## 3. Materials and Methods

The materials were produced using a Vacuum Induction Melting and Casting Systems by PVA TePLA AG company (Wettenberg, Germany), model VSG 002 DS multipurpose system. Initially, 800 g ingots of the master alloy (A0) were cast from the elements of the following purities: Co 99.9%, Cr 99%, Fe 99.99%, Ni 99.99%, Mo 99.95%, and Cu 99.9%. Then the ingot was sectioned into 6 pieces, one being left for the characterization and the other 5 re-melted in an electric arc furnace with a cold copper hearth, where different amounts of Cu and Mo were added. In order to promote homogeneity, each button of approximately 15 mm thickness was re-melted at least 3 times.

Semiquantitative analyses of the alloying elements were performed with a FISCH-ERSCOPE model XUV 773 energy dispersive X-ray fluorescence spectrometer (Helmut Fischer GmbH, Sidelfingen, Germany). This instrument is equipped with a microfocus rhodium X-ray tube, a monotorized XYZ stage, a video microscope for exact positioning of the irradiated area in the sample, and a Si–Pin detector with an energy resolution of <140 eV in terms of full-width at half-maximum for Mn $K_\alpha$ energy. The instrument is

driven by the WinFTM® software (Helmut Fischer GmbH, Sidelfingen, Germany), which is also used for both the spectra acquisition and the treatment. The analyses were carried out at operating conditions of 20 kV using the 1 mm diameter beam collimator and an acquisition time of 50 s. The current was automatically adjusted by the system to reduce the dead time in the detector. The compositions obtained are shown in Table 1, which displays minor differences with the nominal composition. In order to facilitate the referencing through the text, the base alloy is named A0, and the sequential additions of Cu, $Cu_2Mo$, CuMo, $CuMo_2$, and Mo are named $A0|Cu_{10}$, $A0|Cu_7Mo_3$, $A0|Cu_5Mo_5$, $A0|Cu_3Mo_7$, and $A0|Mo_{10}$, respectively, as shown in Table 1.

**Table 1.** Nominal and experimental composition of the alloys characterized. The second column shows the notation employed in the text for the sake of clarity.

| Alloys | Notation | | Composition at. % | | | | | |
|---|---|---|---|---|---|---|---|---|
| | | | Co | Cr | Cu | Fe | Mo | Ni |
| $CoCrFe_2Ni_2$ | A0 | Nominal | 16.67 | 16.67 | - | 33.33 | - | 33.33 |
| | | Experimental | 16.8 | 17.0 | - | 33.4 | - | 32.8 |
| $(A0)_{0.9}(Cu)_{0.1}$ | $A0|Cu_{10}$ | Nominal | 15 | 15 | 10 | 30 | - | 30 |
| | | Experimental | 16.0 | 16.0 | 9.9 | 30.2 | - | 29.5 |
| $(A0)_{0.9}(Cu_2Mo)_{0.1}$ | $A0|Cu_7Mo_3$ | Nominal | 15 | 15 | 6.67 | 30 | 3.33 | 30 |
| | | Experimental | 14.6 | 14.6 | 6.5 | 31.6 | 3.2 | 29.0 |
| $(A0)_{0.9}(CuMo)_{0.1}$ | $A0|Cu_5Mo_5$ | Nominal | 15 | 15 | 5 | 30 | 5 | 30 |
| | | Experimental | 14.5 | 14.5 | 4.9 | 62.3 | 4.8 | 28.8 |
| $(A0)_{0.9}(CuMo_2)_{0.1}$ | $A0|Cu_3Mo_7$ | Nominal | 15 | 15 | 3.33 | 30 | 6.67 | 30 |
| | | Experimental | 14.3 | 14.3 | 3.1 | 33.1 | 6.7 | 28.6 |
| $(A0)_{0.9}(Mo)_{0.1}$ | $A0|Mo_{10}$ | Nominal | 15 | 15 | - | 30 | 10 | 30 |
| | | Experimental | 14.0 | 14.0 | - | 33.5 | 9.3 | 28.2 |

The samples were homogenized for 5 h at 1200 °C and then hot forged until 5 mm thickness was achieved, corresponding to a 67% thickness reduction. Standard metallographic preparation techniques were applied to the as-cast material and the homogenized and forged samples, and all were inspected using a scanning electron microscope, FEG-SEM Hitachi S4800 (Hitachi Ltd., Chiyoda, Tokyo, Japan) equipped with an energy dispersive spectrometer (Oxford INCA, Oxford Instruments plc., Abington, Oxfordshire, UK). In order to reveal the phases present in the microstructure, X-ray diffraction (XRD) studies were performed in a Bruker AXS D8 diffractometer (Bruker AXDS, GmbH, Karlsruhe, Germany) equipped with a Co X ray tube working at 30 mA and 40 kV, a Goebel mirror optics (Bruker AXDS, GmbH, Karlsruhe, Germany) to obtain a parallel and monochromatic beam, and a LynxEye linear detector (Bruker AXDS, GmbH, Karlsruhe, Germany). The area of interest on the sample surface parallel to the forging direction was selected with the help of a video microscope. Conventional θ–2θ scans were collected over a 2θ range, from 35 to 135° with a step size of 0.01°. The whole XRD patters were refined with version 4.2 of the Rietveld analysis program TOPAS (Bruker AXS, Karlsruhe, Germany), using the crystallographic information of the phases present, obtained from the Pearson's Crystal Structure Database for Inorganic Compounds [47]. The refinement protocol included the determination of the structural parameters (unit cell parameter) and the crystallite size and microstrain from the line broadening of the XRD patterns by the double-Voigt approach. For this goal, the instrumental contribution to peak broadening was to eliminate the use of the profile shape functions of a corundum sample measured under the same conditions.

Electron Probe Micro Analysis (EPMA) was carried out in a JEOL Superprobe JXA-8900 M (JEOL Ltd., Tokyo, Japan) microprobe equipped with a wavelength dispersive spectrometer (WDS) to map areas of 400 μm × 800 μm, using a spatial resolution of 1 μm². For the set-up, a voltage, current, and acquisition time of 20 kV, 100 nA, and 20 ms were employed, respectively.

The magnetization measurements of these alloys were performed using a quantum design MPMS-XL SQUID magnetometer (Quantum Design International, San Diego, CA, USA). The magnetization curves were recorded at room temperature for each heat-treated sample by varying the external applied magnetic field from 0 to 50 kOe in steps of 2 kOe. The system takes two measurements at each incremental field step, and the average data point is noted. The results are very accurate as the SQUID (superconducting quantum interference device) can detect minute variations (of the order of $1 \times 10^{-14}$ kOe) of a sample's response to an applied magnetic field.

A DIL805 A/D high resolution dilatometer (TA instruments, New Castle, DE, USA) was employed to measure the change in length using quartz push-rods on cylindrical dilatometric specimens of the alloys under investigation, in a homogenized condition, during continuous heating conditions. Cylinders of 7 mm length and 4 mm diameter were machined for these experiments. The heat treatment consisted in applying a two-stage heating cycle: heating at 5 °C/s from room temperature to 300 °C (where no microstructural alteration is expected) followed by a constant very low heating rate at 0.05 °C/s until 1100 °C.

Finally, aging treatments were carried out for 8 h on the homogenized samples at 500 °C, 700 °C, and 900 °C in a Carbolite furnace (Carbolite Gero, Neuhausen, Germany), model CTF 12/65/550, in which the usual ceramic tube was replaced by a Kanthal alloy tube (less brittle and can resist high temperature oxidation up to 1250 °C). The heat treatments were performed under a gas argon atmosphere in order to investigate the thermal stability and potential precipitation reactions that might take place in the microstructure of the different alloys.

## 4. Results

In this section a detailed microstructural characterization of the samples is presented. Firstly, the results for the as-cast condition are introduced, followed by the homogenized and forged samples. The microstructures obtained after such homogenization at 1200 °C showed single FCC structures without the presence of secondary phases, from which one can conclude that all the compositions are High Entropy Alloys. Secondary phases or precipitation phenomena may actually occur at medium temperatures and a fast screening is performed with dilatometry, from which some reference temperatures are selected to apply the thermal treatments and promote such potential precipitation.

### 4.1. As-Cast Microstructure

A first insight into the microstructures obtained for the as-cast alloys was performed by scanning electron microscopy (SEM). Figure 2 shows some representative backscattered electron (BSE) images of the alloys: (a) A0, (b) A0|$Cu_{10}$, (c) A0|$Cu_7Mo_3$, (d) A0|$Cu_5Mo_5$, (e) A0|$Cu_3Mo_7$, and (f) A0|$Mo_{10}$ in as-cast condition. Most abundant inclusions correspond to black contrast particles with a size ranging from ~5–10 μm. Due to the high concentrations of Cr and O observed in the energy dispersive spectroscopy (EDS) spectra, they were associated with $Cr_2O_3$ oxides. Additionally, thermodynamic calculations performed in the Cantor alloy also show that the formation of $Cr_2O_3$ is more likely than the formation of other oxides, such as $Fe_2O_3$, $MnO_2$, or CoO [48], in agreement with some observations made during oxidation experiments [49,50]. The introduction of Cr in the manufacturing of the alloys indirectly increases the $O_2$ concentration in the melt. As the rest of alloying elements are characterized by a lower affinity to $O_2$ compared to Cr, the $O_2$ present in the melt results in the formation of Cr oxide inclusions. Some examples of these inclusions can be seen in Figure 2.

As-cast A0 base alloy shows neither precipitates nor secondary phases (except for the above-mentioned inclusions, see Figure 2a), which is in agreement with an X-ray diffraction analysis where a full FCC microstructure is detected. The sample A0|$Cu_{10}$ displays, as expected [11,19–21], some Cu-rich precipitates (Figure 2b). In these systems a Cu-rich phase can be present as pure Cu or in combination with some elements from the matrix [36].

Although, due to the size of these particles (lower than 1 μm), it was not possible to determine accurately their composition from their EDS spectra, their semiquantitative analyses showed average Cu contents higher than 80 at. % (Table 2). Rietveld refinement of the XRD pattern obtained from this material showed the presence of an FCC phase with a lattice parameter of 3.5869 Å in a content of about 10 mass-%. As this value for the lattice parameter was only slightly lower than that reported for the pure Cu, it was concluded that the amount of this element in solid solution was low, but also that other elements from the matrix were present in the Cu-rich precipitates, in accordance with the EDS spectra measurements. The samples A0|$Cu_7Mo_3$ (Figure 2b) and A0|$Cu_5Mo_5$ surprisingly did not show any Cu-rich precipitates, despite still having a considerable amount of Cu. The X-ray diffraction patterns agreed with the SEM observation and a single FCC was detected in both samples. The fact that Cu does not segregate or precipitate in A0|$Cu_7Mo_3$ and A0|$Cu_5Mo_5$ alloys cannot be attributed to the difference in entropy with respect to A0|$Cu_{10}$. The configurational entropy of A0|$Cu_{10}$ is $-1.52R$, while $-1.58R$ and $-1.59R$ correspond to A0|$Cu_7Mo_3$ and A0|$Cu_5Mo_5$, respectively. Neither the magnetic contribution to the entropy is expected to be different, as it is described in Section 4.2.2. Anyway, the presence of some nano-sized Cu-rich particles cannot be excluded using the experimental techniques employed in this work.

The A0|$Cu_3Mo_7$ (Figure 2e) and A0|$Mo_{10}$ (Figure 2f) alloys clearly showed a dendritic structure, where the clear-contrast areas corresponded to the Mo-rich regions, a probable consequence of the large dissimilarities of Mo with respect to the other elements in the composition. These Mo-rich areas were contained inside the Mo-rich precipitates, which were identified as $Fe_7Mo_6$ (μ phase) from the XRD analysis of the A0|$Mo_{10}$ sample. Mo-rich particles were present in both the A0|$Cu_7Mo_3$ and the A0|$Mo_{10}$ samples, but while in the A0|$Mo_{10}$ alloy such precipitates were large and homogenously distributed along the interdendritic regions (Figure 2), their presence was very scarce and residual in the A0|$Cu_7Mo_3$ alloy (Figure 2d). Rietveld refinement of the XRD pattern of the A0|$Mo_{10}$ sample confirmed the presence of the μ phase. However, in the Rietveld analysis performed considering a structural model consisting of an austenite and a μ phase, the difference plot showed asymmetric broadening of the austenite peaks that could not be associated with cell size and/or the microstrain effect. This broadening is associated with two FCC structures with slightly different lattice parameters. Dendritic solidification causes some elements (such as Mo) to concentrate in the inter-dendritic regions, and the differences in the alloying element between the dendritic and the interdendritic areas led to two different values for the austenite lattice constant.

When the microstructures present in A0|$Cu_7Mo_3$, A0|$Cu_5Mo_5$ and A0|$Cu_3Mo_7$ alloys were compared with those present in the A0|$Cu_{10}$ and A0|$Mo_{10}$ alloys, we concluded that the combination of Cu and Mo inhibited or reduced dramatically the precipitation kinetics of both the Cu-rich and the $Fe_7Mo_6$ particles. It is remarkable that two elements, Cu and Mo, which promote precipitation phenomena in the base alloy when added individually, collaborate to avoid precipitation when added simultaneously, without any significant increase in the entropy. In addition to this, Cu and Mo have a very positive mixing enthalpy [40].

The compositional EDS micro-analysis of the different precipitates described above is given in Table 2. The Cu-rich precipitates are estimated to be mostly composed of Cu (~80 at. %) and display an FCC structure following the X-ray diffraction analysis, in accordance with previous works on similar systems [11,19–21]. The Mo-rich precipitates were identified as the μ phase, having a $D8_5$ $Mo_6Fe_7$ phase crystal structure with the cell parameters $a = 4.7196$ Å and $c = 25.6131$ Å. A similar result has been observed in other works from the literature [5,51,52]. Their composition, as shown in Table 2, agrees with the stoichiometry observed, with the participation of other elements from the matrix and with the special importance of Cr, which is not surprising, as this phase appears in the Fe-Cr-Mo system at medium and elevated temperatures [53].

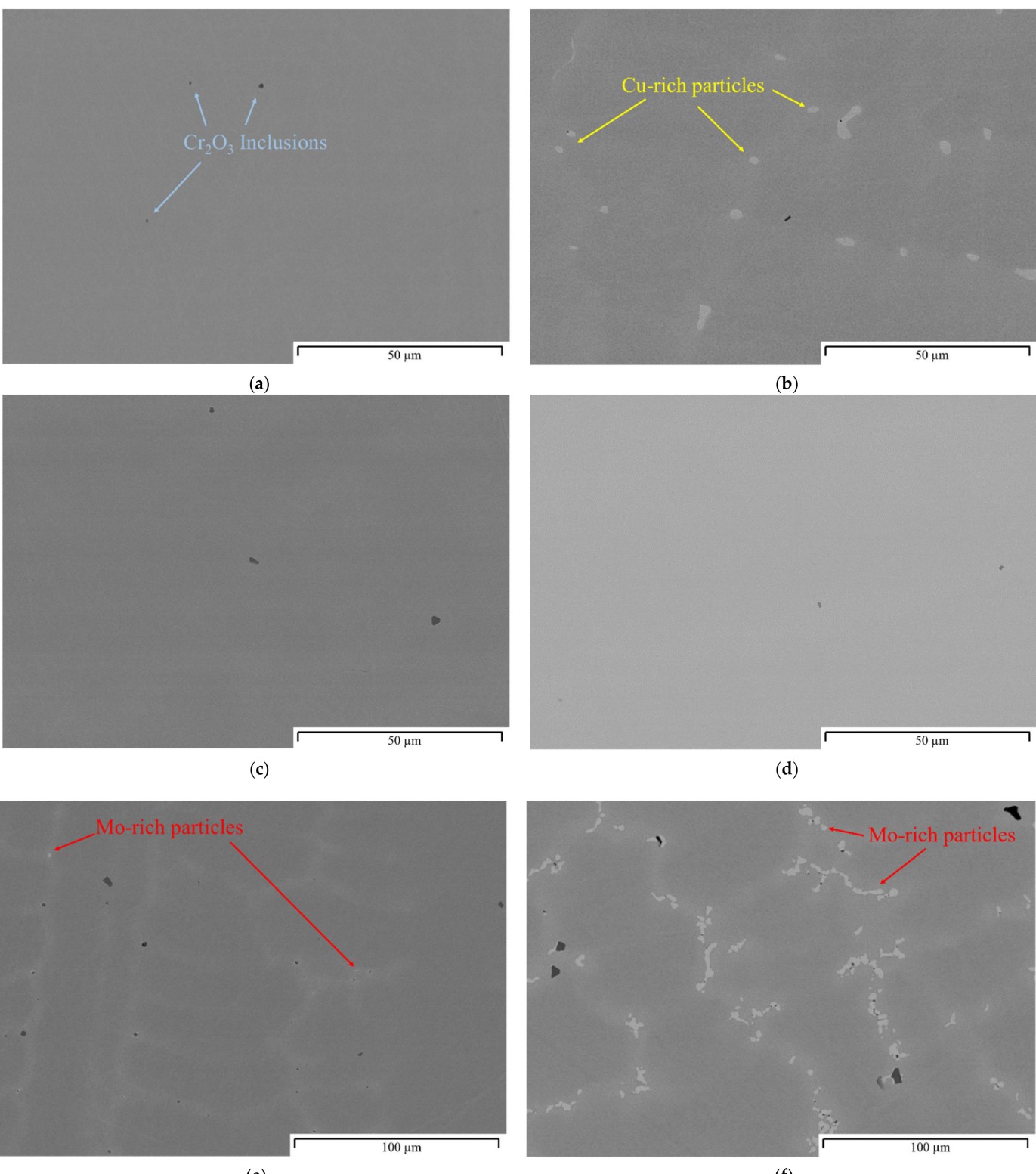

**Figure 2.** Backscattered electron (BSE) images in as-cast condition of the alloys: (**a**) A0, (**b**) A0|Cu₁₀, (**c**) A0|Cu₇Mo₃, (**d**) A0|Cu₅Mo₅, (**e**) A0|Cu₃Mo₇, and (**f**) A0|Mo₁₀. The dispersed dark contrast spots correspond to Cr-oxides resulting from impurities in the raw material or contamination during alloy production.

**Table 2.** Composition of the secondary phases observed in the as-cast condition obtained by energy dispersive spectroscopy (EDS) microanalysis.

| | Composition at. % | | | | | |
|---|---|---|---|---|---|---|
| | **Co** | **Cr** | **Cu** | **Fe** | **Mo** | **Ni** |
| Cu-rich precipitates | 2.5–3.0 | 1.5–2.5 | 80–82 | 5–6 | - | 7–8 |
| Mo-rich precipitates | 9–12 | 15–20 | - | 18–23 | 32–38 | 12–17 |

*4.2. Homogenization at 1200 °C and Forged Microstructures*

4.2.1. X-ray Diffraction

X-ray diffraction patterns of the samples after the homogenization treatment at 1200 °C for 5 h and the hot forging are shown in Figure 3a. In these patterns, only the presence of an FCC crystal structure is observed, whose diffraction peaks move to the lower 2θ values as the Mo content is increased, as shown in Figure 3b for the (220) reflection. The evolution of the lattice parameter, as calculated by the Rietveld refinement of the XRD patterns, is shown in Figure 3c, together with the cell parameter calculated following the method described in [54], where an averaged lattice quadratic potential is proposed by means of the unit cell parameters and the bulk modulus of the constitutive elements. The equilibrium of such potential corresponds to the lattice parameter of the alloy, and it has been calculated assuming that all the alloying elements are in a single FCC solid solution. A good correlation is observed between the calculated and experimental unit cell parameters. As the nominal radius of Mo when joined to other atoms by metallic bonds (1.39 Å) is considerably bigger than the metallic radius of the rest of the alloying elements (which range between 1.24 and 1.28 Å [55]), the incorporation of Mo into the FCC will produce an increase in its lattice parameter, as observed in Figure 3c.

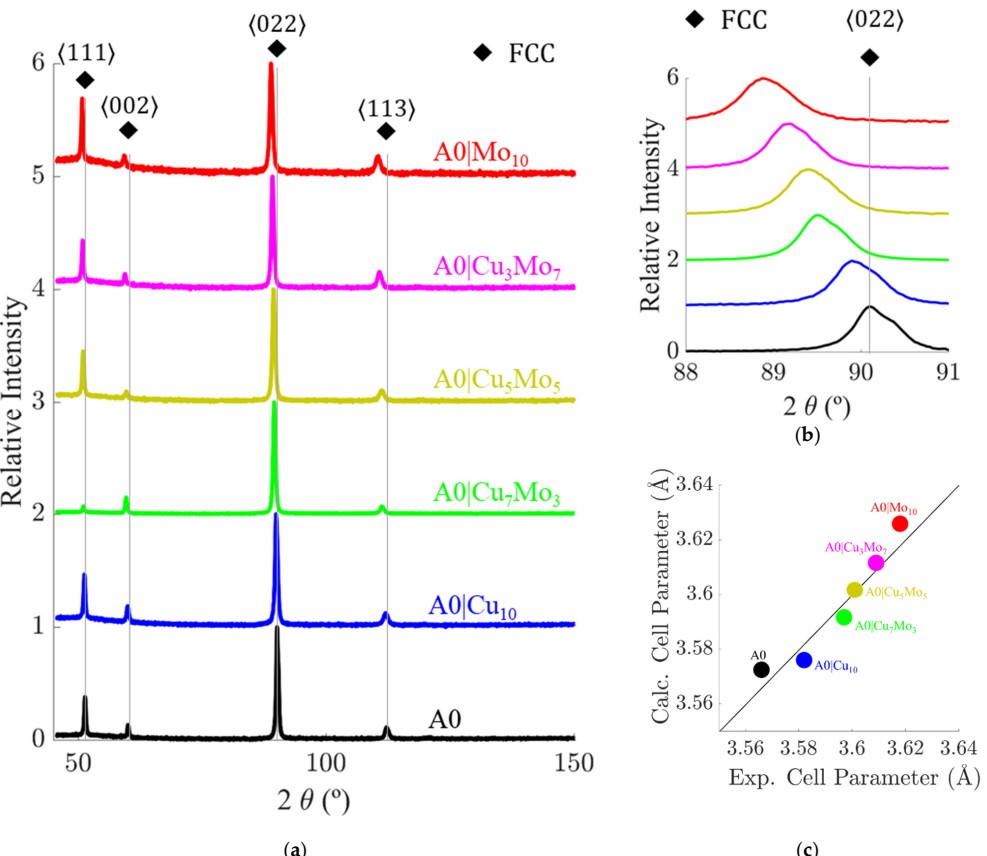

**Figure 3.** (**a**) X-ray diffraction patterns of the homogenized and forged samples, (**b**) detail of X-ray diffraction peak, (**c**) experimental vs. calculated cell parameter of the alloys.

### 4.2.2. Saturation Magnetization Measurements

Thermodynamics depends strongly on the magnetic behavior of alloys [56,57], and it also has a great influence on the stability of the microstructure, not only via its contribution to the enthalpy, but also to the entropy [10,58], with the addition of terms accounting for the variation of magnetic moment per atom in the lattice [59].

Thus, a screening of the magnetic behavior at room temperature was performed in the samples via magnetic measurements as a function of an applied field of up to 50 kOe. The obtained hysteresis cycles can be observed in Figure 4a. This figure shows that with the base alloy A0 together with A0|Cu$_{10}$, both achieve a maximum magnetization and average magnetic moments per atom of 0.53 and 0.43 µb, respectively. This small difference between both alloys can be attributed to the decrease in the content of the elements with a stronger contribution to the magnetism of the alloy (Fe, Co and Cr). Those are not large magnetic moments (just for the reference, pure BCC–Fe has 2.22 $\mu_b$ [60]), but they are indeed not negligible. This figure also shows that the Mo solute additions tend to reduce the saturation moment of the alloys. As all the alloys from A0|Cu$_{10}$ to A0|Mo$_{10}$ have the same content of Fe, Co, and Cr, it was concluded that the possible magnetic contribution of these elements tends to be strongly modified by the presence of Mo. The reduction in the average magnetic moment per atom with the Mo content can be seen in Figure 4b.

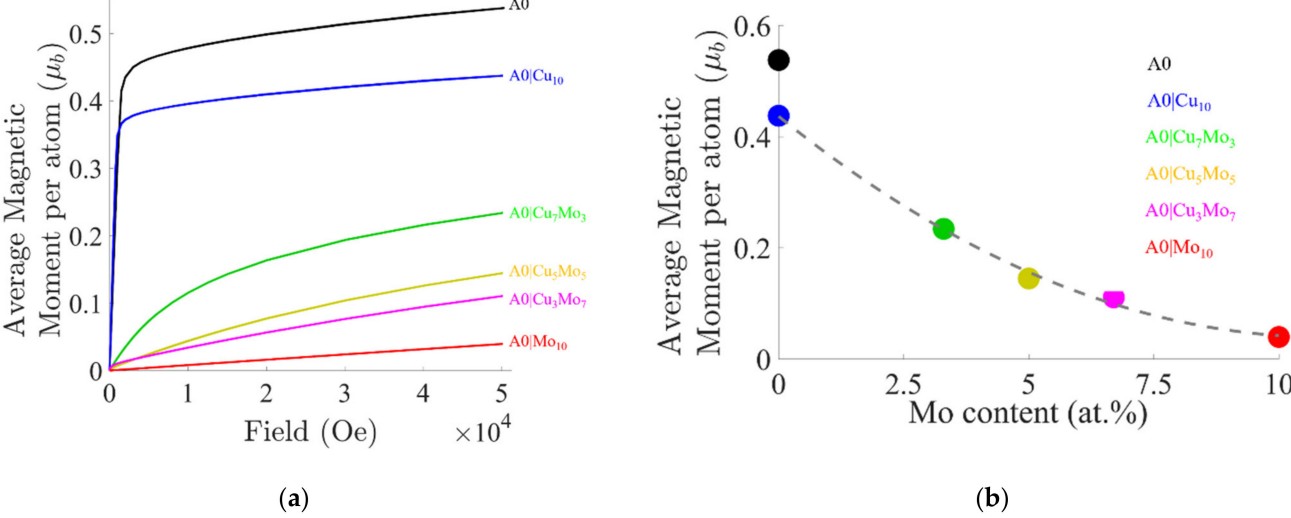

(**a**)          (**b**)

**Figure 4.** (**a**) Average magnetic moment per atom of the samples, (**b**) evolution of the average magnetic moment per atom with respect to Mo content.

### 4.2.3. EPMA: Electron Probe Micro Analysis

The EPMA maps for Co, Cr, Cu, Fe, Mo, and Ni for the homogenized samples are shown in Figure 5. The maps have been performed in steps of 1 µm$^2$ and cover an area of 400 × 800 µm$^2$, making 320,000 points measured per sample. These intensity maps are translated into composition and mathematically (numerically) treated to minimize the noise induced by the measurement, as was performed in previous works [61,62]. Appendix A contains a detailed explanation of the filter applied. This noise, intrinsic to the raw data, is responsible for a variation in total composition of between 95 at. % to 105 at. % in some points. The filter applied functions by smoothing the compositional maps obtained at different orders until the sum of the elemental compositions at each point measured is below a certain threshold, close to 100 at. %. This smoothing has been applied with order 6, which provides a sufficient filtering to highlight the compositional variations in the material. The original data obtained can also be seen in Appendix A.

The results show that the compositional variation across the microstructure does not exceed 1 at. %. The elements Co, Cr, and Ni are homogeneously distributed in the microstructure, whereas the Fe, Cu, and Mo have a special interaction and show some

heterogeneities in the material, always below the 1 at. % variation. It is surprising that Cu and Mo tend to segregate together at the expense of Fe, rather than Fe and Mo at the expense of Cu, as the enthalpy of mixing would predict. This highlights the different behavior of the elements in complex systems such as HEAs, where the results in binary systems do not explain their behavior when they interact with many other elements in solid solution. That does not mean necessarily that the information in low order systems cannot be used and lacks interest for HEAs, but special attention must be paid to its application, as suggested previously [63].

Of special importance is the stabilizing effect of the combination of Cu and Mo. This is unveiled later when different samples with different concentrations of Cu/Mo are compared. In this regard, in the alloy A0|Cu$_{10}$, the Cu tends to segregate more severely than in A0|Cu$_7$Mo$_3$, A0|Cu$_5$Mo$_5$, and A0|Cu$_3$Mo$_7$. Similarly, in the alloy A0|Mo$_{10}$, in the absence of Cu, the Mo also tends to segregate more severely than in A0|Cu$_7$Mo$_3$, A0|Cu$_5$Mo$_5$, and A0|Cu$_3$Mo$_7$. This fact could be related to the formation of Cu-rich precipitates in A0|Cu$_{10}$ as well as Mo-rich precipitates in A0|Mo$_{10}$. In other words, such precipitations are enhanced in the areas with a higher concentration of Cu or Mo, acting as preferential regions for precipitation. Similarly, the precipitation is retarded in the A0|Cu$_7$Mo$_3$, A0|Cu$_5$Mo$_5$, and A0|Cu$_3$Mo$_7$ alloys as the preferential regions are fewer or too small for nucleation.

In order to highlight the relationship or interaction dynamics among the elements as well as their homogeneity in the microstructure, a series of scatter plots of the compositions for each pair of elements can be seen in Figure 6. Each plot represents the overall content of two elements from the EPMA analysis in the corresponding $400 \times 800$ μm$^2$ scanned area. Every point of every plot represents the composition of two such elements for each step (1 μm$^2$) scanned. In order to lighten the figure, the axes marks have been avoided, and the length of each axis is $\pm 1$ at. % around the nominal composition of each element, as was shown in Table 1.

Several interesting conclusions can be extracted from this figure. On the one hand, there are conclusions which concern the compositional variability of each element and, on the other hand, there are those which concern the relationship between different elements. Concerning the compositional variability, it is observed that Fe has the largest compositional variation, followed by Ni and Co with medium variability, while Cr has a very low variability. These four elements keep this behavior in all the samples, irrespective of the Cu and Mo content. It is worth noting here that none of these four elements has any special correlation with the others, with the exception of a positive correlation between Co and Fe, which is especially pronounced in the A0|Cu$_5$Mo$_5$ sample. In other words, these four elements display an excellent ability to mix in solid solution as they avoid segregation or areas where different phases could coexist. On the other hand, both the Cu and the Mo have a rather different behavior, which leads to the formation of Cu-rich precipitates, as was already highlighted in the BSE images (Figure 2a). Cu has a large compositional variation in A0|Cu$_{10}$ (where no Mo exists), which decreases with the addition of Mo. Similarly, Mo has a large compositional variation in A0|Mo$_{10}$ (where no Cu exists), leading to the formation of segregation patterns and precipitation of Mo-rich phases, as was unveiled in the BSE images (Figure 2d). This behavior decreases with the addition of Cu. Two groups of elements compose the observed compositional correlations: Cu-Mo, which shows positive correlation, as well as Co-Fe. Logically negative correlations arise from the crossed interactions of Cu-Fe, Co-Cu, Fe-Mo, and Co-Mo.

This is surprising behavior if it is assumed that binary systems behave similarly in multicomponent systems. This assumption is the basis of using CALPHAD or even some of the solid solution rules based on thermodynamics. Under this premise, positive enthalpies would provide negative correlations, and negative enthalpies would follow positive correlations. This does not occur for the Cu-Mo interaction, both with positive correlation and enthalpy of mixing, while it does occur for the Cu-Fe and the Cu-Co, both with negative correlation and positive enthalpies. This result highlights the necessity

of strengthening the thermodynamic behavior in multicomponent systems in order to predict their behavior.

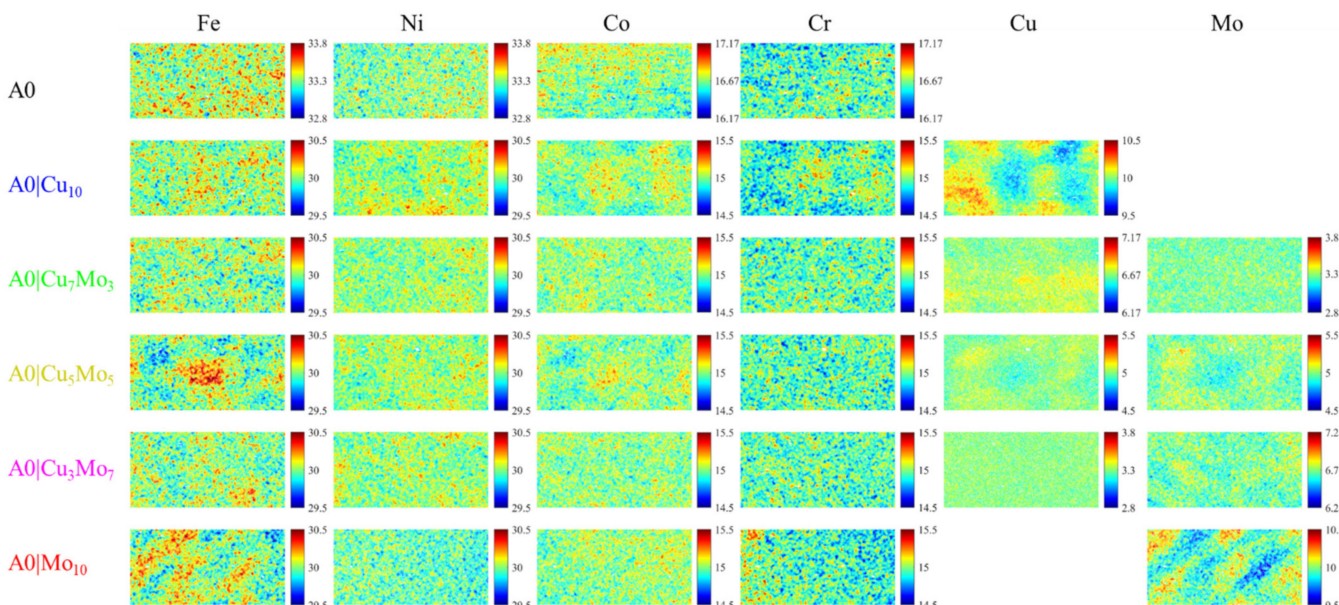

**Figure 5.** Electron Probe Microanalysis (EPMA) maps of the homogenized and hot-forged samples. The color bars indicate compositional variations in at. %. For the sake of simplicity and comparison purposes, all the color bars have a variation of 1 at. %. The size of the maps analyzed is 400 × 800 μm$^2$.

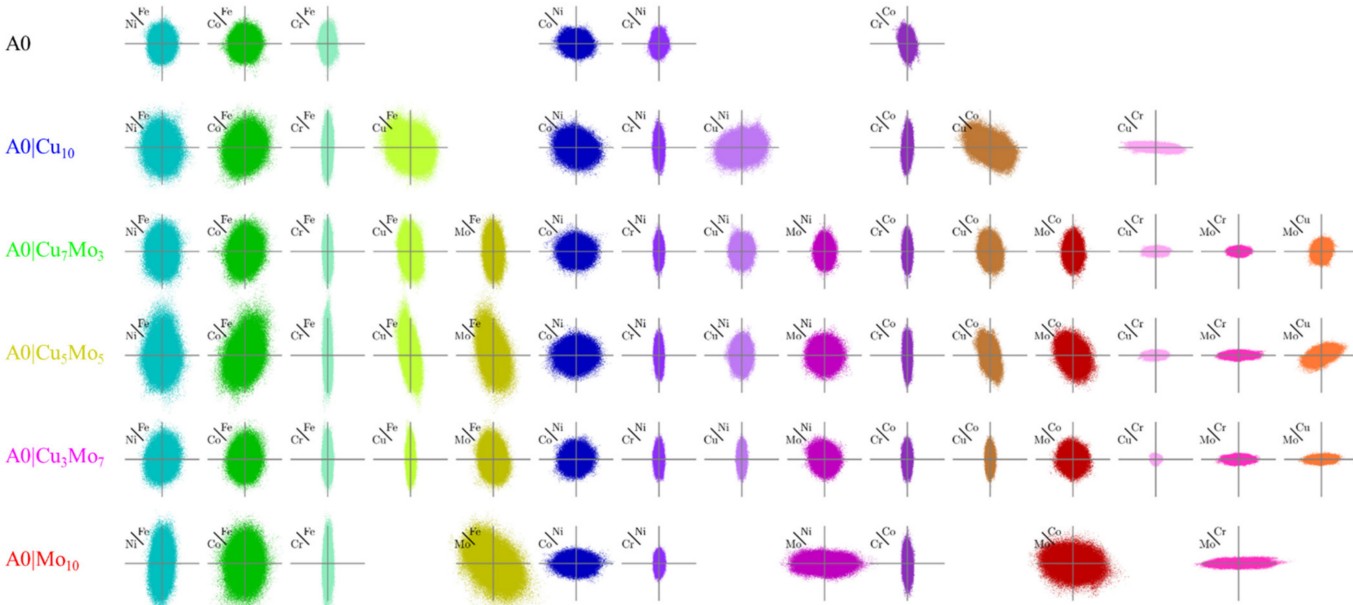

**Figure 6.** Correlation of pairs of elements for all samples based on the EPMA analysis. The length of each axis represents 1 at. % compositional variation around the content of each element.

The main conclusion from the analysis of the EPMA is that the samples A0|Cu$_7$Mo$_3$ and A0|Cu$_5$Mo$_5$ are the most homogenous alloys and that the presence of Cu and Mo may help to stabilize the FCC crystal structure as Mo may avoid the formation of Cu-rich regions. Neither Cu nor Mo modifies the excellent behavior of the base alloy as no special interaction occurs with Fe, Ni, Co, or Cr.

*4.3. Microstructure after Heat Treatments*

From the characterization results obtained after the homogenization and hot-forging of the alloys, it can be concluded that all the proposed alloys can be categorized as High Entropy Alloys. Nevertheless, such classification could indeed be reserved for alloys displaying a single phase at all temperatures at equilibrium, but this actually rarely happens. Proof of this is that the Cantor alloy, identified as among the few real High Entropy Alloys, may develop MnNi phases, Cr-rich phases, and Fe–Co phases when subjected to severe plastic deformation, increasing dramatically the number of grain boundaries, thus facilitating the diffusion of the elements and creating multiple nucleation sites [64]. Even more revealing is that such a phase decomposition occurs at temperatures as low as 450 °C after 5 min of heat treatment, in the case of MnNi and Cr-rich phases, while the FeCo phase needed much longer exposure times for the same temperature. That does not discredit the HEAs in the literature as the main characteristic to be compared to conventional alloys is the large microstructural stability over a wide temperature range. This effect actually enlarges at high temperatures with the entropic contribution to the Gibbs free energy, as opposed to many other alloys.

This section explores the thermal stability of the six alloy compositions. Firstly, a fast screening is carried out with high resolution dilatometry, applying a slow continuous heating, to identify at which temperatures the precipitation phenomena may be taking place; secondly, isothermal heat treatments are performed at different selected temperatures, and finally, the characterization of the resulting microstructures is presented.

4.3.1. Dilatometry

High resolution dilatometry has been employed extensively in the past to characterize solid–solid phase transformations in different types of steels [65,66]. It permits the identification of microstructural changes with great accuracy. This is possible, provided that these microstructural changes involve phases that have different specific atomic volumes, resulting in the overall change of the alloy sample volume. Although when it comes to the investigation of precipitation reactions in metals and, particularly, in steels, differential scanning calorimetry (DSC) has been generally proven to be a very sensitive technique [67–71]; high resolution dilatometry has also found an application in the detecting of this type of phase changes in the precipitation of hardening steels [72–74].

The temperature evolution of the relative change in the length ($\Delta L/L_0$, with $L_0$ the initial length) curves of the six alloys are displayed in Figure 7a. A positive offset has been applied to the curves for the sake of clarity. The curves show that the alloys do not undergo any mayor microstructural changes that can be associated with the precipitation of the secondary phases. A smooth increase occurs without abrupt changes for all alloys. Therefore, this suggests that their matrices keep in the FCC crystal structure from room temperature until 1100 °C. There are nevertheless some slight differences in the behavior of the curves. To unveil these differences, the coefficient of linear thermal expansion (CLTE), a derivative of the relative change in length, has been calculated from the curves shown in Figure 7a and displayed in Figure 7b. Similarly, the curves are displayed with a positive offset for a better visualization. The possible precipitation phenomena occurring during heating can be identified in this way. Note that, as described in Section 3, a fast heating of the sample is induced until 300 °C to speed the test. No attention must be then paid to the CLTE for the low temperature range as the test is too fast and the temperature too low to allow any microstructural change. The peaks observed around 300 °C in all the samples are due to the change in the heating rate. Because precipitation reactions are thermally activated processes, they will be promoted by the application of low heating rates; therefore, the focus should be put on the slow heating rate stage of these curves (T > 300 °C)

From Figure 7b, the first conclusion is that the A0 base alloy shows a quasi-linear increase with temperature with a very weak wavy behavior, especially when compared to the other alloys. This suggests that no precipitation occurs during heating and that

the microstructure remains stable. A similar result was found in the equimolar CoCr-FeNi alloy [18,75]. The CoCrFeNi alloy is actually compositionally far from the current CoCrFe$_2$Ni$_2$, from a conventional metallurgical point of view. This would confirm the large stability of the CoCrFeNi system, where large variations in the composition remain unaltered in their thermal stability and would support further investigations on non-equiatomic CoCrFeNi composition, as in the CoCrFe$_2$Ni$_2$ alloy considered here.

The alloy A0 | Cu$_{10}$ has a similar behavior until 700 °C, but a prominent increase (peak) can be seen in Figure 7b between 700 °C and 1000 °C. This could be associated with the precipitation of Cu-rich phases, as observed in [19,20] for a Co$_{22}$Cr$_{22}$Fe$_{22}$Ni$_{22}$Cu$_{11}$ alloy, which seems to precipitate around 500 °C, together with a Cr-rich FCC phase. The presence of a Cu-rich phase is explained in these works based on the positive enthalpy of the mixing of Fe-Cu, Co-Cu, Ni-Cu, and Cr-Cu. The composition composed of A0 | Cu$_{10}$ alloy seems then more stable. This system precipitation phenomenon occurs at higher temperatures, even though there is an increase in Fe content with respect to the Co$_{22}$Cr$_{22}$Fe$_{22}$Ni$_{22}$Cu$_{11}$ alloy (the largest positive enthalpy is on the Fe-Cu interaction), and a decrease of the mixing entropy ($\Delta S_{mix}$ = 1.58$R$ in the Co$_{22}$Cr$_{22}$Fe$_{22}$Ni$_{22}$Cu$_{11}$ and $\Delta S_{mix}$ = 1.52$R$ in the A0 | Cu$_{10}$, where $R$ is the ideal gas constant).

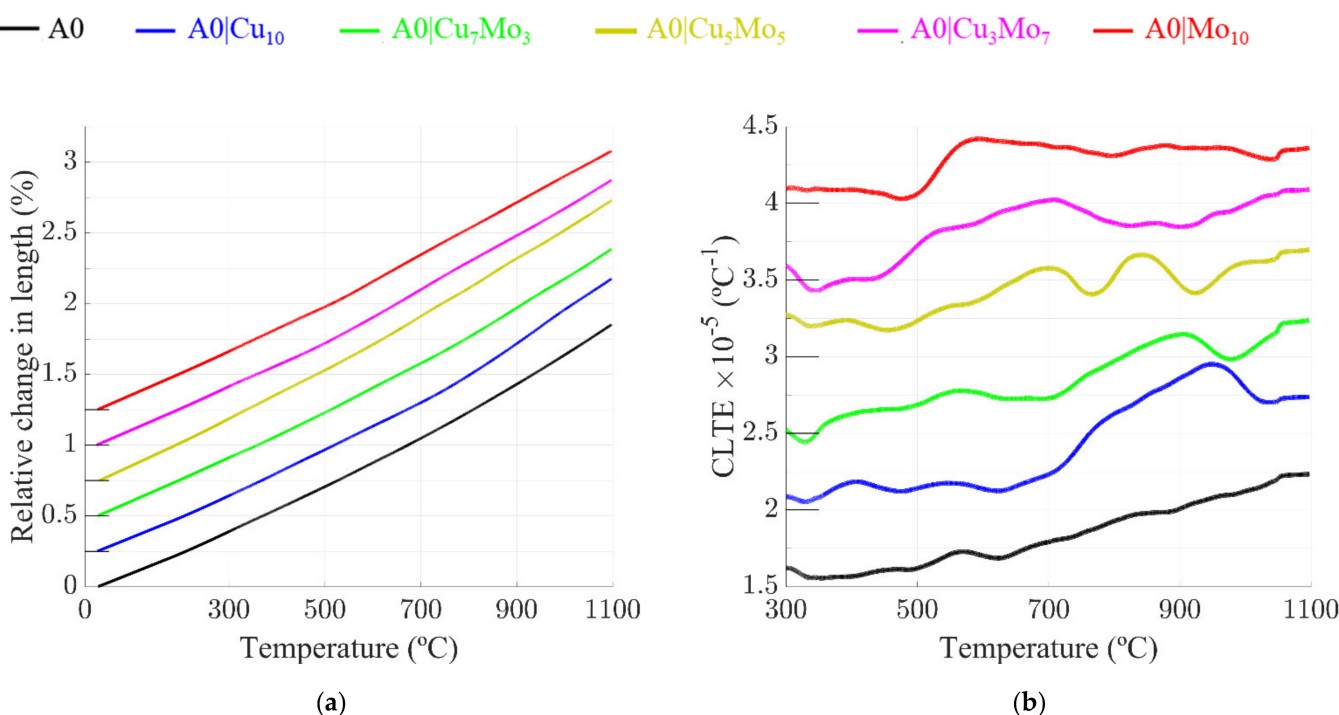

**Figure 7.** Dilatometry curves of the alloys A0–A5. (**a**) Relative change in length $\Delta L/L_0$ (%) vs. temperature, (**b**) CTLE (°C$^{-1}$) (Linear thermal expansion coefficient). The curves are displayed with an offset where the corresponding origins are marked with small horizontal lines.

The alloy A0 | Cu$_7$Mo$_3$ also shows a peak in the same temperature range as that displayed for A0 | Cu$_{10}$ but is much less pronounced. This is the first evidence that the possible Cu-rich precipitate formation in the CoCuCrFeNi system may be inhibited or reduced by the presence of Mo. This is surprising for two reasons. Firstly, the Mo content is very low and little effect would be expected on the microstructural stability of the alloy, and secondly, the enthalpy of the Mo-Cu interaction is even larger than the other Cu interactions, which should promote an even more severe Cu segregation.

On the other hand, the alloys A0 | Cu$_3$Mo$_7$ and A0 | Mo$_{10}$ show a single peak at medium temperatures, around 600–700 °C. A previous work showed the precipitation of the $\mu$ and $\sigma$ phases in the CoCrFeNiMo$_x$ (x = 0, 0.3, 0.5, 0.85) alloys, although the possible temperatures at which these secondary phases may form is not reported [5]. Nevertheless,

only the $\mu$ phase was observed in the XRD patterns of these alloys after heat treatments (as it can be seen later on Section 4.3.3). Both phases form at medium and high temperatures, but the $\mu$ phase has a larger stability at medium temperatures [76] and the temperature formation fits reasonably well with the $\mu$ phase present at 700 °C in the Cr-Fe-Mo ternary system [53].

Finally, the A0|Cu$_5$Mo$_5$ alloy seems to have a mixture of the behavior of the Cu precipitation of A0|Cu$_{10}$ and the $\mu$ precipitates in A0|Cu$_3$Mo$_7$ and A0|Mo$_{10}$, showing a transition between both compositional extremes.

4.3.2. Heat Treatments at 500 °C, 700 °C and 900 °C

As a result of the dilatometry analysis and to investigate further what phases could be forming at high temperatures in the FCC matrix of these alloys, three temperatures were selected to perform isothermal ageing treatment for 8 h. This heat treatment duration is considered long enough to promote the formation of the phases that were potentially forming during slow heating at 0.05 °C/s (Figure 7). It is worth noting that the HEAs have been postulated to have low diffusivity [77], which could affect the heat treatment times to induce precipitation, if this should occur at equilibrium. Nevertheless, other more recent works have suggested doubts about this property, especially as a general rule for HEAs [78,79]. The heat treatment conditions investigated are considered long enough to provide an insight on the potential thermal stability of these alloys.

As was discussed in the previous section two main peaks associated with the possible formation of Cu-rich and Mo-rich phases were observed around 900 °C and 700 °C, respectively, in the CLTE vs. temperature plot (Figure 7b). For these reasons, samples from all the alloys were aged at these two temperatures. Additionally, a third temperature was selected (500 °C) to investigate the thermal stability at lower temperatures where weaker peaks have been observed in the CLTE vs temperature plot.

As could be expected from the dilatometry results, the alloy A0 showed no precipitation in the microstructure after the heat treatments at the three temperatures. This highlights the large stability of this system and leads to the conclusion that the weak wavy behavior observed in the temperature variation of the CLTE may be just due to the instrumental noise of the technique. Thus, the A0 sample output could be taken as the reference result that should be expected in the absence of precipitation. The alloy A0|Cu$_{10}$ showed, instead, a small amount of Cu-rich precipitates decorating the grain boundaries only in the case of 900 °C, which agrees well with the results displayed in Figure 7b. The small amount of Cu-nano precipitates in the matrix at lower temperatures cannot be discarded, but their volume fraction must be much lower than those precipitated at 900 °C at the grain boundaries as a change in the curve would be detected at such low temperatures. Figure 8a shows a SEM image of the microstructure obtained at such a temperature. Their presence is not important, but indeed it cannot be ignored. It could be the triggering of a most pronounced grain boundary precipitation for longer times. A similar result was found in [20], in a system of CoCrFeNi + Cu$_{0.5}$, but the precipitation started at 500 °C (with minor evidence of Cu-rich precipitates at 350 °C). This marks a large difference with the A0|Cu$_{10}$ alloy used here, where no precipitation was found even at 700 °C. As a comparison, the entropy in the current alloy A0|Cu$_{10}$ (1.5218 $R$) is lower than in the CoCrFeNi + Cu$_{0.5}$ (1.5811 $R$) alloy. This means that the entropy is not the only parameter stabilizing the microstructure until 700 °C in the A0|Cu$_{10}$ alloy. Non-equiatomic compositions have demonstrated in the past to possess sometimes better microstructural stability than equiatomic compositions [6], even though these latter ones have higher entropy. A proper selection of the composition facilitates the stabilizing task of the entropy, and it allows expanding the temperature ranges at which a single solid solution exists. It is also worth noting that in [20] the treatment was maintained for 24 h, and not only 8 h as in the current case, which has to be taken into account for comparison purposes. Nevertheless, it is assumed here that 8 h should be enough time to precipitate such Cu-rich precipitates as these easily form.

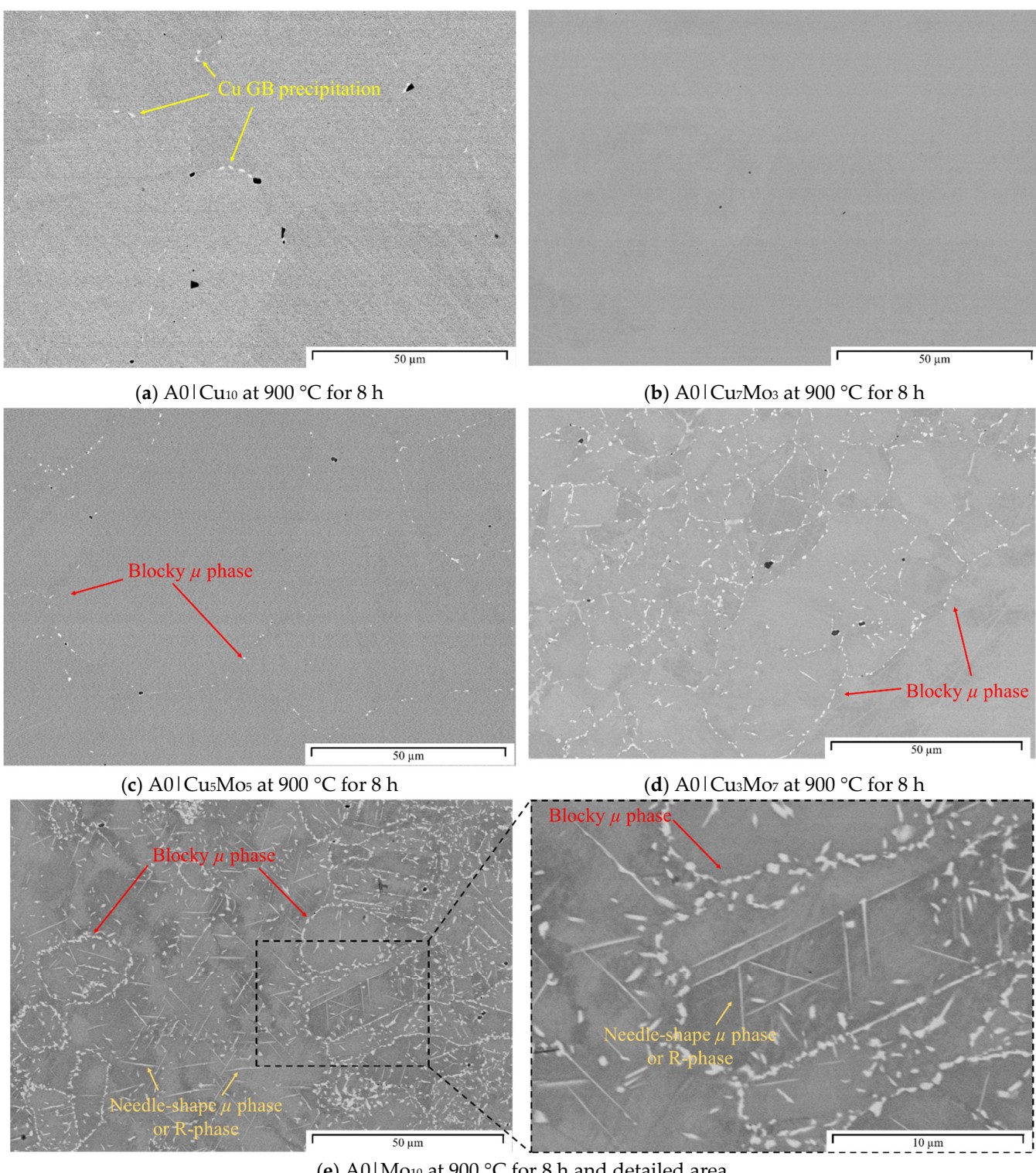

**Figure 8.** Backscattered electron (BSE) images of some treated samples at 900 °C. (**a**) A0|Cu$_{10}$, (**b**) A0|Cu$_7$Mo$_3$, (**c**) A0|Cu$_5$Mo$_5$, (**d**) A0|Cu$_3$Mo$_7$, (**e**) A0|Mo$_{10}$ with detail of Mo-rich precipitates and needle-shape precipitates. All alloys have been heat treated for 8 h. GB stands for grain boundary. The dispersed dark contrast spots correspond to O- and S-inclusion resulting from impurities in the raw material or contamination during alloy production.

The alloy A0|Cu$_7$Mo$_3$ shows a different microstructure with respect to A0|Cu$_{10}$. It does not show any coarse Cu-rich precipitation at the grain boundaries. It seems that Cu remains in solid solution even at 900 °C, though transmission electron microscopy should be employed to discard the nanoprecipitation of Cu-rich precipitates. At the increasing of

the Mo content, the microstructure of the alloy A0 | $Cu_5Mo_5$ shows a similar behavior to that of the A0 | $Cu_7Mo_3$ alloy at 500 °C and 700 °C. The difference arises at 900 °C with the formation of Mo-rich precipitates, as can be seen in Figure 8c, similarly to the precipitates present in the as-cast material. Nevertheless, the Mo-rich precipitates in the as-cast material were present in the A0 | $Cu_3Mo_7$ and A0 | $Mo_{10}$ alloys but not in the A0 | $Cu_5Mo_5$ alloy. In this case, the Mo-rich precipitates nucleate preferably at the grain boundaries. The forged microstructure offers in this case many more nucleation sites for precipitation as opposed to the as-cast material, which could explain the difference observed. A similar behavior has been observed in the case of the alloy A0 | $Cu_3Mo_7$ where the behavior at 500 °C and 700 °C is similar to that of the A0 | $Cu_5Mo_5$ alloy. As in the A0 | $Cu_5Mo_5$ case, in the current one Mo-rich precipitates form at the grain boundaries, fully decorating their microstructure. This can be observed in Figure 8d.

Finally, the alloy A0 | $Mo_{10}$, without Cu, shows that the Mo-rich precipitates start their nucleation at the grain boundaries at the 700 °C treatment. This suggests that Cu retarded the nucleation of the Mo-rich precipitates in the A0 | $Cu_5Mo_5$ and A0 | $Cu_3Mo_7$ alloys, in which no nucleation was observed at 700 °C. No other reason can be attributed to this lack of nucleation in A0 | $Cu_5Mo_5$ and A0 | $Cu_3Mo_7$ as in both cases the grain boundaries remained free to allow nucleation. Additionally, such nucleation is inhibited or retarded for the A0 | $Cu_7Mo_3$. This effect is symmetrical to the one described above with the inhibiting effect of Mo over Cu precipitation. Both Cu and Mo collaborate to inhibit Mo- and Cu- rich precipitate nucleation at the grain boundaries, respectively.

For the case of A0 | $Mo_{10}$ at 900 °C, needle-shape or acicular precipitates grow from the grain boundaries and inside the grains. They can be observed in Figure 8e inside the grains, which are at the same time decorated by Mo-rich precipitates. They can be as long as the full length of a grain and keep a clear crystallographic relationship. The angle of such precipitates inside a grain is constant, but each grain has a different angle relationship, which is consequence of the different alignment with respect to the observation plane. A similar result is found in [53] where needle/thin-plate-like precipitates are described as R-phase metastable precipitates which appear around 800–900 °C are described as needle/thin-plate-like. Nevertheless, the observed $\mu$ phase detected previously in the grain boundaries has two different morphologies [80]. Firstly, a block-like $\mu$, similar to the one observed in the grain boundaries and secondly, a needle-shaped $\mu$ which evolves from a fiber-like structure. Their composition varies slightly, with a higher content of Mo for the block-like $\mu$.

### 4.3.3. X-ray Analysis after Heat Treatment of 900 °C

X-ray analysis was carried out in all the samples treated at 900 °C. The corresponding XRD patterns can be observed in Figure 9a. Concerning the matrix, the samples display a single FCC structure with the exception of A0 | $Cu_{10}$ and A0 | $Mo_{10}$ where two FCC structures are detected. For the case of A0 | $Cu_{10}$, a Cu-rich FCC is observed with a larger cell parameter with respect to the matrix, but lower than pure Cu. In the A0 | $Mo_{10}$ alloy, two FCC solid solutions were detected, where the difference was attributed to a slight variation in the alloying elements (specially Mo) between the dendritic and interdendritic areas. The energy dispersive spectroscopy (EDS) micro-analysis carried out in different areas shows variations between 7–11 at. % in Mo across the matrix, whereas the other elements do not show significant variations.

With respect to the formation of intermetallic phases, only A0 | $Cu_3Mo_7$ and A0 | $Mo_{10}$ showed a 0.45% and 2.62% mass fraction of the $\mu$ phase, respectively. The A0 | $Cu_5Mo_5$ did not show any secondary phase within the detection limits of this technique, although some traces are seen in the backscattered electron images shown in Figure 8c. As opposed to the results shown in [5,52] in the CoCrFeNi + Mo systems, no $\sigma$ phase was detected in the alloys and only the $\mu$ phase was observed, as found in [51] in the same system.

The cell parameter variation is displayed in Figure 9b, again with respect to the theoretical calculations assuming a fully FCC solid-solution structure, as in Figure 3c,

where the two FCC phases for the $A0|Cu_{10}$ and $A0|Mo_{10}$ are displayed accordantly. The Cu-rich phase measured cell parameter, compared to the pure Cu cell parameter, shows a lower value, suggesting the presence of other alloying elements, in agreement with the EDS spectra performed in the particles of this phase. The two FCC solid solutions seen in the $A0|Mo_{10}$ alloy display a small but not negligible difference in the cell parameter, due mainly to the different Mo content.

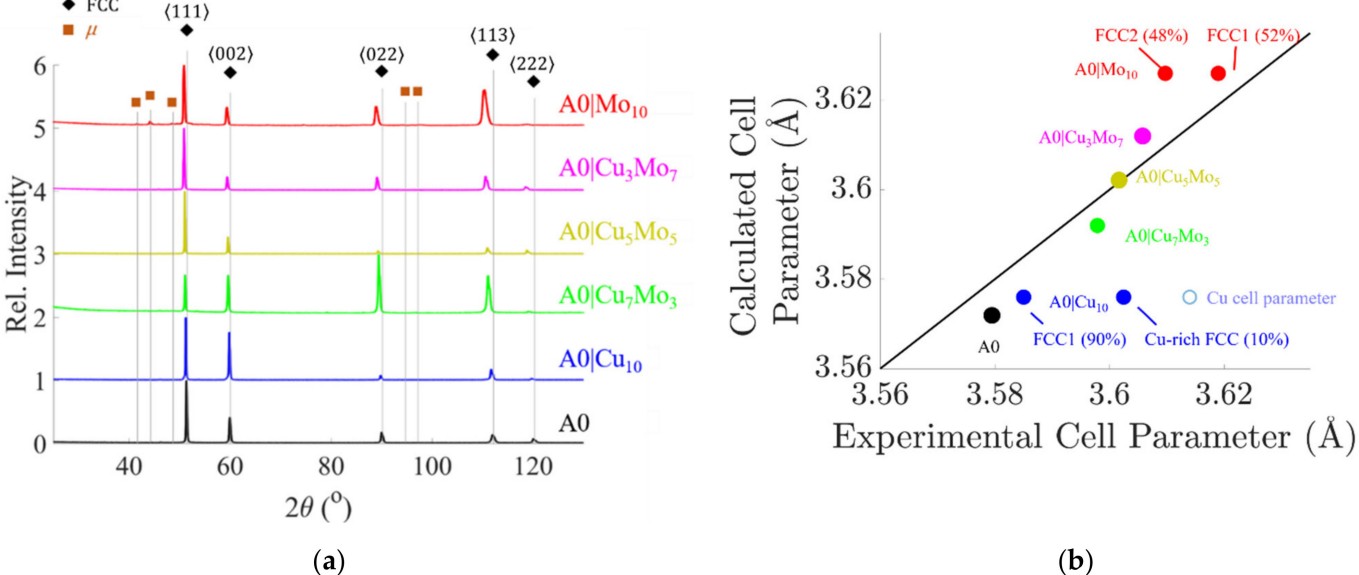

(a) (b)

**Figure 9.** (**a**) X-ray diffraction patterns of samples treated at 900 °C for 8 h and (**b**) comparison between the measured and computed cell parameters.

After this analysis, Table 3 summarizes the microstructures obtained by means of the different heat treatments, as described previously. As can be observed, the A0 and $A0|Cu_7Mo_3$ alloys showed a large microstructural stability, while the $A0|Cu_{10}$ alloy displayed some Cu-rich precipitates at 900 °C. This small addition of Mo on the Cu-containing alloys seems to have had a strong stabilizing effect on the microstructure.

**Table 3.** Summary of the heat treatments applied to the alloys and the precipitates obtained. All alloys have a face centred cubic (FCC) crystal structure, and the additional precipitates are indicated where they correspond. The matrix (denoted with -) is a solid-solution FCC structure. FCC1 + FCC2 stands for two different FCC structures, and GB stands for precipitates observed at the grain boundaries.

| Treatment | A0 | $A0|Cu_{10}$ | $A0|Cu_7Mo_3$ | $A0|Cu_5Mo_5$ | $A0|Cu_3Mo_7$ | $A0|Mo_{10}$ |
|---|---|---|---|---|---|---|
| 500 °C—8 h | - | - | - | - | - | - |
| 700 °C—8 h | - | - | - | - | - | Blocky μ (GB) |
| 900 °—8 h | - | FCC Cu-rich (GB) | - | -<br>Traces of blocky μ (GB) | -<br>Blocky μ (GB) | FCC1 + FCC2<br>+<br>Blocky μ (GB)<br>+<br>Needle-shape μ<br>or R phase |
| 1200 °C—5 h (homogenization treatment) | - | - | - | - | - | - |

On the other hand, the alloys with a larger Mo content showed Mo-rich precipitation at high temperatures in the grain boundaries and a needle-shape μ phase or R phase in the case without Cu ($A0|Mo_{10}$). The absence of any peak corresponding to the R phase in the XRD patterns suggests that the acicular precipitation observed corresponds to the needle-shape μ phase. Nevertheless, more advanced techniques, such as transmission electron microscopy, are needed to confirm this point. Irrespective of this, we conclude that Cu also has an important effect on Mo-containing alloys as it inhibits or retards the

formation of Mo-rich precipitates and improves at the same time the formation of a single FCC phase matrix.

## 5. Summary and Conclusions

(a)  Based on solid-solution formation rules, a segregation parameter ($\lambda$), and thermo-dynamic calculations, the $CoCrFe_2Ni_2$ system was developed by adding up to 10 at. % of Cu, Mo, or the combination of both elements. The alloys designed showed a single FCC phase once the microstructure was homogenized by a thermal treatment at 1200 °C, and therefore they constitute a new family of HEAs.

(b)  The EPMA compositional maps revealed that the interdendritic areas present an enrichment of Cu and Mo accompanied by a noticeable Fe depletion. This result is unexpected as thermodynamics predicts that these two elements should reject each other, in a similar way to Cu with Fe, and the Mo-enriched areas should be also enriched by Fe due to the positive enthalpy of the Fe-Cu interaction. This Fe depletion from the Mo-enriched zones highlights the complexity of the atomic interactions in HEAs and the difficulty of their prediction.

(c)  The non-equimolar composition has shown an improvement in the system stability, delaying the appearance of secondary phases, especially for the Cu-containing alloys, in comparison with the behavior of some equimolar Cu-containing compositions described in the literature.

(d)  Cu-Mo interaction is responsible for inhibiting or delaying the precipitation of both Cu-rich and Mo-rich particles in the A0|$Cu_7Mo_3$, A0|$Cu_5Mo_5$, and A0|$Cu_3Mo_7$ alloys, and the maintenance of a single FCC crystal in the A0|$Cu_7Mo_3$ alloy during aging treatments. In A0|$Cu_5Mo_5$, A0|$Cu_3Mo_7$, and A0|$Mo_{10}$ alloys, Mo-rich blocky precipitates of the $\mu$ phase decorate the grain boundaries after an aging heat treatment at high temperatures, where the diffusion of the elements is faster. In the case of the A0|$Cu_5Mo_5$ and A0|$Cu_3Mo_7$ alloys, the precipitation occurs only at 900 °C, while for the A0|$Mo_{10}$ the precipitation had already started at 700 °C due to the absence of Cu that can inhibit or retard this phenomenon. For this alloy, aging at 900 °C also produces the formation of acicular precipitates, probably corresponding to the needle-shape $\mu$ phase.

**Author Contributions:** Conceptualization, I.T.-C. and D.S.-M.; methodology, I.T.-C. and D.S.-M.; software, I.T.-C.; validation, I.T.-C. and D.S.-M.; formal analysis, I.T.-C., D.S.-M. and J.A.J.; investigation, I.T.-C. and D.S.-M.; resources, S.M.; data curation, I.T.-C.; writing—original draft preparation, I.T.-C.; writing—review and editing, D.S.-M., J.A.J. and S.M.; visualization, I.T.-C., D.S.-M. and J.J.-A.; supervision, I.T.-C. and D.S.-M.; project administration, I.T.-C., D.S.-M., J.A.J. and S.M.; funding acquisition, I.T.-C., D.S.-M., J.A.J. and S.M. All authors have read and agreed to the published version of the manuscript.

**Funding:** This research was funded by the Regional Government of Madrid for support via the DIMMAT Programme (S2013/MIT-2775) and Mat4.0 project (S2018/NMT-4381).

**Institutional Review Board Statement:** Not applicable.

**Informed Consent Statement:** Not applicable.

**Data Availability Statement:** The data presented in this study are available on request from the corresponding author.

**Acknowledgments:** I. Toda-Caraballo is grateful for the financial support of the fellowship 2016-T2/IND-1693, from the Programme Atracción de talento investigador (Consejería de Educación, Juventud y Deporte, Comunidad de Madrid). The authors are also grateful to the following laboratories and technicians from CENIM-CSIC: Phase transformations (Miguel Acedo Ojeda), X-ray diffraction (Irene Llorente), and Electron Microscopy (Antonio Tomás Lopez and Martin Ian Maher) and for the provision of laboratory facilities and technical support. Alfredo Fernández Larios from the Centro Nacional de Microscopıa Electronica (CNME), Complutense University (UCM), is acknowledged for the provision of the EPMA facility and for the technical support. Julio Romero

de Paz from the Research Support Center of Physical Techniques (CAI-Técnicas Físicas, UCM) is acknowledged for the provision of the SQUID magnetometer facility and the technical support.

**Conflicts of Interest:** The authors declare no conflict of interest. The funders had no role in the design of the study; in the collection, analyses, or interpretation of data; in the writing of the manuscript; or in the decision to publish the results.

## Appendix A

The raw data from the EPMA experiments can be seen in Figure A1. This figure displays the measured composition without applying any filter to the original data. Although no special compositional variations are observed in different areas, with the exception of Cu in A1 and Mo in A5, the apparent compositional variation of each element varies up to 4 at. %. A particular feature of the raw data is that the addition of all the elements, representing the total composition, which should give a homogenous 100 at. %, displays a variation larger than 10%. This means that some areas will have 105 at. %, while others 95 at. %. The rule applied in this work is that the variation of the total composition should not be larger than 1 at. %. This rule has been chosen without a specific physical meaning, but it is a reasonable value which allows displaying the compositional variations across the microstructure with a minimum noise in the data.

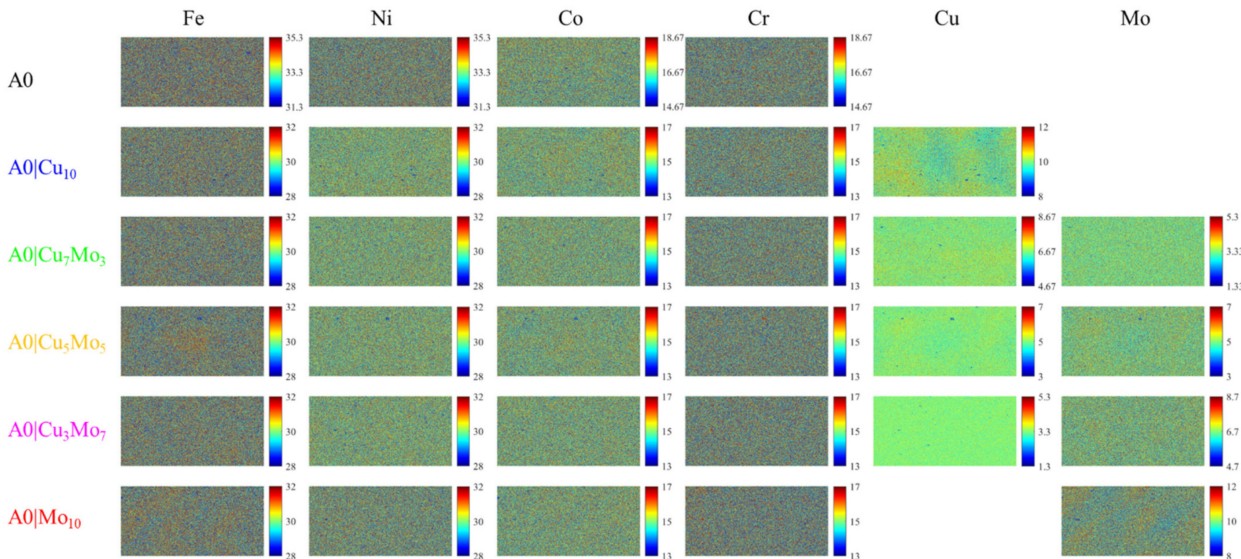

**Figure A1.** Untreated EPMA maps of the homogenized samples. The color bars indicate the variation of each elemental content in at. %. For the sake of simplicity and comparison purposes, all the color bars have a variation of 4 at. %.

We can conclude that by using this filter the analysis of the results is dramatically different with respect to that of using the raw data. The filter allows the observation of the many different interactions that occur in homogenous solid solutions.

The filter methodology is very simple. The intensities measured from the EPMA are represented by a matrix of 400 rows and 800 columns. Let $M$ be one these matrices and $M_s$ be the smoothed matrix:

$$M_s(i,j) = \frac{1}{(2n+1)^2} \sum_{k=i-n}^{k=i+n} \sum_{l=j-n}^{l=j+n} M(l,k), \text{ for } i,j \geq n+1 \tag{A1}$$

where $n$ is the order of the filter. Note that the size of the matrix $M_s$ is $(i-n) \times (j-n)$ rather than the size of $M$, which is $i \times j$. This represents a small loss compared with the benefit of minimizing the noise. This method simply modifies the value of one element of the matrix by the average value of $\pm n$ elements around it. The algorithm is straightforward,

and it is of easy implementation. The filter employed in this work has been of order 6, and it is the minimum value that provides a noise of the overall composition below 1 at. %.

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
