# Peer review of "Microstructural Stability of the CoCrFe2Ni2 High Entropy Alloys with Additions of Cu and Mo"

_metals, doi:10.3390/met11121994_

Round 1

Reviewer 1 Report

Nice work. I enjoyed reading. Major revisions are in order for the authors to address the comments below:

“Some examples are solid solution hardening [7], precipitation hardening [8] or even exploring TRIP/TWIP effects [9] , as well as many other properties such us oxidation [10], corrosion [11], or even improvements in irradiation resistance [12], without excessive compromise of its microstructural stability”: the processability of these alloys has also been studied recently. See for example 10.1016/j.scriptamat.2021.114219 and 10.1016/j.matdes.2020.108505 and revise.

“. Of special relevance in this system is the observation of deformation twining [9], demonstrating that HEAs could benefit from this hardening effect”: other deformation mechanisms exist for this alloy system.

“not surprising then that more than 25 works can be found in the literature for the AlCoCrFeNi with different Al contents.”: way more than 25.

“Cu nevertheless easily segregates forming Cu-rich precipitates”: this is also true for Cu in other HEA systems. See for example 10.1007/s11661-019-05564-8 and further complement the introduction.

“leading to unexpected behaviors that may affect from corrosion to magnetism.”: please detail these behaviors.

“due to its large FCC stability and”: quantify?

“The results are depicted in Error! Reference source not found.”: check this!

“of Cu, Mo, and Cu+Mo in different contents”: by how much?

“achieved and determine the compositional window for obtaining stable single solid solu-”: only in the as-cast condition or after heat treatment?

“limited to 15 at.% so that oxidation protection is obtained, but low enough to minimiz”: have the authors tried above 15% to see if sigma is expected to form? Below 9 at% of Cr the corrosion resistance would likely be low.

“The results at allowing variations of Fe, Ni and C”: where are these results?

“ions can be found in Error! Reference source not found..”: again, careful with this!

How did the authors confirmed the inclusions (for example Cr2O3)? EDS? XRD? Clarify please. High so much oxide being formed? This will likely reduce the corrosion properties of the alloy as the Cr is being depleted from the matrix.

“he cell parameter calculated following the method described”: what was the method? Clarify.

“treated to minimize the noise induced by the measuremen”: how was this done? Clarify.

For fig 5: do the authors have an SEM image of the analyzed regions? To see the correlation between the composition and microstructure features.

“and µ precipitates in”: were these precipitates indexed in the XRD?

Did the authors evaluated the hardness of the alloys? This would be nice to correlate with the microstructure features.

Author Response

Nice work. I enjoyed reading. Major revisions are in order for the authors to address the comments below:

“Some examples are solid solution hardening [7], precipitation hardening [8] or even exploring TRIP/TWIP effects [9] , as well as many other properties such us oxidation [10], corrosion [11], or even improvements in irradiation resistance [12], without excessive compromise of its microstructural stability”: the processability of these alloys has also been studied recently. See for example 10.1016/j.scriptamat.2021.114219 and 10.1016/j.matdes.2020.108505 and revise.

Response: The processability has been also cited, and the references added.

“. Of special relevance in this system is the observation of deformation twining [9], demonstrating that HEAs could benefit from this hardening effect”: other deformation mechanisms exist for this alloy system.

Response: Agree with that. The sentence is only intended to show some special features of this system, as for the case of adding Al (next paragraph) where the solid solution hardening effect and phase transformation to BCC depending on Al content are cited, but these are not the only particularities of this system. This introduction makes a short review of the main families, without the intention to go deep in each of them. The authors believe that such analysis is out of the scope of this work and it would extend unnecessarily the length of the manuscript. For further information about the Cantor system (matter of this comment), several references were originally added, including a complete review where more information is included.

“not surprising then that more than 25 works can be found in the literature for the AlCoCrFeNi with different Al contents.”: way more than 25.

Response: The sentence has been change into the following text to avoid inaccuracies, but still highlighting the interest in this system “. It is not surprising then that this system is among the most studied in the literature.”

“Cu nevertheless easily segregates forming Cu-rich precipitates”: this is also true for Cu in other HEA systems. See for example 10.1007/s11661-019-05564-8 and further complement the introduction.

Response: Some references have been added to include the Cu segregation in other families. The new sentence is as follows:

“Cu nevertheless easily segregates forming Cu-rich precipitates in this system [11], [17]–[19] and others, as in the AlCoCrFeNi+Cu [20]–[22], the AlCoCrFeNiTi+Cu [23] or the AlCoNiTiZn+Cu [24] families. This highlights the inclination of this element to elemental separation, which occurs specially at the grain boundaries, often inducing a detrimental effect into the mechanical properties or even into the microstructural stability due to the matrix compositional variations”

“leading to unexpected behaviors that may affect from corrosion to magnetism.”: please detail these behaviors.

Response: Concerning the corrosion, the behaviour has been cited and reference, while the magnetism has been removed. The sentence is now as follows:

 “The thermodynamics of the alloy may vary significantly from the original solid solution alloy [35] leading, for instance, to localised corrosion [11].”

“due to its large FCC stability and”: quantify?

Response: A short sentence and corresponding reference has been added.

“due to its large microstructural stability, with FCC phase structures remaining unchanged in temperature ranges between 350 ºC and 1350 ºC [18]”

“The results are depicted in Error! Reference source not found.”: check this!

Response: In the original word document, the reference remains correct, although the pdf version showed the error. After reviewing the build pdf, we have noticed that this reference error appears several times, and the authors apology for this.  The revised version will remove cross checks to avoid this processing issue.

“of Cu, Mo, and Cu+Mo in different contents”: by how much?

Response: The additions are shown in Table 1. In order to avoid repetitive information, the sentence now refers directly to such table

“additions of Cu, Mo, and Cu+Mo in different contents as shown in Table 1 provided below”

“achieved and determine the compositional window for obtaining stable single solid solu-”: only in the as-cast condition or after heat treatment?

Response: Both conditions are considered in this work. It is worth noting that in the HEAs literature, many alloys are considered HEAs (single solid solution) in the homogenized condition (after heat treatment of 1200 ºC, for instance), without further investigation at lower temperature ranges. It is the intention, in this work, to investigate the microstructural stability of this family at different temperatures. The sentence has been modified accordantly

achieved and determine the compositional window for obtaining stable single solid solutions, after the homogenization but also after thermal treatments at lower temperatures”

“limited to 15 at.% so that oxidation protection is obtained, but low enough to minimiz”: have the authors tried above 15% to see if sigma is expected to form? Below 9 at% of Cr the corrosion resistance would likely be low.

Response: The thermodynamic calculations showed that there is still some room for increasing Cr above 15 at. % in the base alloy (considering Cr, Fe, Ni and Co only), but the presence of Mo quickly induces sigma precipitation. Nevertheless, after considering the reviewer comment, the authors agree that the sentence could be misleading, as the intention is to limit Cr to 15 at.% so that Cr is always higher than 15 at.% to obtain corrosion resistance, and not the opposite, which would be to limit Cr so that its content is 15 at.% maximum. The sentence has been changed accordantly to facilitate the reader understanding of the alloy design method.

In the calculations, the Cr content is required to be higher than 15 at.% to guarantee that oxidation protection is obtained, but low enough to minimize the σ-phase formation.”

“The results at allowing variations of Fe, Ni and C”: where are these results?

Response: The large amount of calculations cannot be displayed efficiently in the manuscript, while the main conclusion is that the Cu-segregation is predicted to be very similar irrespective of the base alloy, within the proposed limits. The authors believe that showing computed phase diagrams in a wide compositional window will not contribute to the manuscript, while it will use unnecessary space.  Nevertheless, a sentence has been added to summarize and quantify the results obtained.

“The results at allowing variations of Fe, Ni and Co show that the predicted segregation of Cu (secondary Cu-rich FCC phase) behaves quite similarly irrespective of the base alloy content. A Cu-content of 10 at.% is predicted to induce the presence of a Cu-rich FCC structure in temperatures around 1000-1100 ºC, while such temperature decreases down to 600-700 ºC for lower Cu-contents (around 3 at.%). This occurs in all base compositions, excepting for cases with high content of Co, where a substantial increase in the formation temperature of the Cu-segregation occurs, even reaching the solidus temperature.”

“ions can be found in Error! Reference source not found..”: again, careful with this!

Response: Same answer as above.

How did the authors confirmed the inclusions (for example Cr2O3)? EDS? XRD? Clarify please. High so much oxide being formed? This will likely reduce the corrosion properties of the alloy as the Cr is being depleted from the matrix.

Response: The Cr2O3 inclusions where characterized during SEM analysis by performing EDS microanalysis on the larger particles and their composition characterized. Clearly, the inclusions did not show any other element in their chemical composition. The Cr content measured also suggested that the oxide formed is chromia. Their morphology and persistence after homogenization were also clear indications of their nature. Their formation is related to the presence of some O in the original Cr material employed. Thermodynamic calculations performed in the Cantor alloy have also shown that the formation of Cr2O3 is more likely than the formation of other oxides like Fe2O3, MnO2 or CoO [1]. Some observations made during oxidation experiments support this prediction [1-3]. These new references have been added into the manuscript in section 4.1

“Additionally, thermodynamic calculations performed in the Cantor alloy have also shown that the formation of Cr2O3 is more likely than the formation of other oxides like Fe2O3, MnO2 or CoO [49], in agreement with some observations made during oxidation experiments [50], [51].”

The authors agree that the corrosion and oxidation behaviour can be influenced for a depletion of Cr in the matrix, but the EDS performed in the matrix several times per sample analysed showed not substantial compositional variation of Cr in the matrix. Such variations were similar as the ones detected in the other elements, and were considered usual minor compositional variations in the matrix (in any case, lower than 1 at.%). Additionally, as the nominal Cr content is much higher the minimum required to obtain corrosion protection, any minor depletion of Cr due to the Cr2O3 presence will be still in compositional ranges with enough Cr to provide protection.

[1] L Wang et al.. Corrosion Science, 167, 2020, pp.108507.

[2] J. DÄ…browa, et al, Oxidation of Metals (2021) 96:307–321.

[3] R. Gordo, et al, JOM (2015), 67, 2326–2339, 

“he cell parameter calculated following the method described”: what was the method? Clarify.

Response: a sentence has been added to clarify the method, as requested.

“…the cell parameter calculated following the method described in [48], where an averaged lattice quadratic potential is proposed by means of the unit cell parameters and bulk modulus of the constitutive elements. The equilibrium of such potential corresponds to the lattice parameter of the alloy, and it has been calculated assuming that all alloying elements are in a single FCC solid solution”

“treated to minimize the noise induced by the measuremen”: how was this done? Clarify.

Response: The explanation of the method is in Appendix A. It was originally referred later, but it is now cited after this sentence to state clearly where the numerical treatment is detailed.

“Such intensity maps are translated into composition, and mathematically (numerically) treated to minimize the noise induced by the measurement, as it was done in previous works [55], [56]. Appendix A contains a detailed explanation of the filter applied.”

For fig 5: do the authors have an SEM image of the analyzed regions? To see the correlation between the composition and microstructure features.

Response: Unfortunately, we do not have the corresponding SEM images for the EPMA maps, but we agree that it is a good suggestion for further communications.

“and µ precipitates in”: were these precipitates indexed in the XRD?

Response: Yes, the precipitates were indexed and are shown later when the microstructures of the thermal treatments are reported. Figure 9 shows the XRD at 900 ºC (such figure is cited in the revised version in this part of the manuscript), to support that the precipitation observed during dilatometry measurements can be attributed to µ precipitates, in accordance also to previous works in the literature cited in the text.

Did the authors evaluated the hardness of the alloys? This would be nice to correlate with the microstructure features.

Response:  Unfortunately, hardness measurements have not been performed and cannot be reported.

Reviewer 2 Report

I have reviewed the submission on the microstructure stability of the CoCrFe2Ni2 HEA with Cu and Mo additions. The title is of interest for the readers; however, there are some issues that need further explanation or amendment as follows:

1) The major question is the author tried to age the HEAs for 8 h to examine if there is a ppt formed within the structure. According to the literature and the sluggish diffusion nature of multicomponent alloys, 8h would be low enough to decide regarding this issue and extended time is expected for such observations. This item needs to be addressed carefully.

2) Microstructure characterization is not well carried out and it is not acceptable that the authors claim one ppt could be A or B! they have to clearly characterize the phase using advanced techniques such as DPs in TEM. Using XRD might be not useful for characterization.

3) Fig.2 is not really useful and cannot convey any significant issue.

4) More papers on the addition of Cu into the chemistry of HEAs needed to be reviewed to address the effect of the addition of elements that have positive mixing enthalpy. Reviewing the following papers could be helpful:

[a] Materials Science and Engineering: A 825, 141875

[b] Materials Letters 272, 127866

Author Response

I have reviewed the submission on the microstructure stability of the CoCrFe2Ni2 HEA with Cu and Mo additions. The title is of interest for the readers; however, there are some issues that need further explanation or amendment as follows:

1) The major question is the author tried to age the HEAs for 8 h to examine if there is a ppt formed within the structure. According to the literature and the sluggish diffusion nature of multicomponent alloys, 8h would be low enough to decide regarding this issue and extended time is expected for such observations. This item needs to be addressed carefully.

Response: The sluggish diffusion in HEAs has been postulated initially as an important feature of these alloys. Some works have been analysing such effect [1] , and compare with other alloys, finding a lower diffusion coefficient. Nevertheless, other more recent works [2,3] have suggested doubts about this property, especially as a general rule for HEAs.

We agree that longer thermal treatments could promote or enhanced the observed precipitation phenomena in this and other systems. The targeted temperatures for structural applications are trying to push the operating temperatures to the range between 700-800 ºC; for this reason, the proposed testing conditions (900 ºC – 8h) were intended to enhance the ageing and, thus, accelerate the precipitation reactions. These reactions would, otherwise, take place at a slower pace at lower temperatures (nucleation rate increases exponentially as the temperature increases). Thus, the heat treatment conditions investigated are considered long enough to have a taste of the potential thermal stability of these alloys. In any case, the authors agree with the reviewer that longer times would be of great interest and additional experiments are now being performed in these alloys at lower temperatures (700 ºC) for times up to 3 months to complement the observation made at 900 ºC.

[1]  K-Y. Tsai et al, Acta Materialia, 61 (2013), 4887-4897

[2] S. V. Divinski, Diffusion Foundations, 17 (2018) 69–104

[3] A. Mehta, Materials Researh Letters 9 (2021) 239-246)

A sentence in the section 4.3.2, after defining the heat treatment conditions, has been added to address the point raised by the reviewer:

“It is worth noting that the HEAs has been postulated to have low diffusivity [78], which could affect the heat treatment times to induce precipitation, if this should occur at equilibrium. Nevertheless, other more recent works have suggested doubts about this property, especially as general rule for HEAs [79], [80]. The heat treatment conditions investigated are considered long enough to have an insight on the potential thermal stability of theses alloys.”

2) Microstructure characterization is not well carried out and it is not acceptable that the authors claim one ppt could be A or B! they have to clearly characterize the phase using advanced techniques such as DPs in TEM. Using XRD might be not useful for characterization.

Response: The authors agree that other advanced techniques (TEM or also APT) could have been included in the work, which always provide additional information. Nevertheless, we believe that the current characterization which uses several techniques, is enough to describe with substantial accuracy the precipitates found. While it is true that TEM would provide a definite proof that the observed precipitates are μ-precipitates and not σ-phase nor R-phase, the X-ray analysis did not detect neither of these phases, while it detected clearly μ-precipitation. Such precipitates correlate well with the SEM analysis and the corresponding microanalysis performed. Additionally, both results are in accordance with the cited previous works in the literature where μ-precipitates have been found Nevertheless, in the manuscript it is already suggested that TEM would be necessary for their full characterization (page 21, line 527) , and that the results obtained here are only a strong indication of the precipitation of  μ-precipitates, while R-phase has not been totally discarded, as shown in Table 3.

3) Fig.2 is not really useful and cannot convey any significant issue.

Response: Figure 2 shows the as-cast materials, prior to any homogenization or heat treatment. The intention is to show the potential microstructural stability of these alloys and their microstructure in as-cast condition, where the base alloy A0 as well as A0|Cu7Mo3 and A0|Cu5Mo5 samples showed no precipitation or secondary phases.

The materials have been subjected later to homogeneization, showing no particular microstructural features since they are full solid solutions (in accordance to the thermodynamic calculations) and therefore the corresponding micrographs would not provide any significant added value in this case. The corresponding X-ray diffraction analysis and EPMA measurements have been included in these cases as a description of their microstructure.

Later, the microstructural features occurring after thermal treatments are described by analysing the SEM images and the corresponding figures are shown in section 4.

With this sequence of different results and corresponding figures at different stages of their treatments, the authors believe that the microstructures are sufficiently described, and therefore we kindly suggest to keep Figure 2 to provide a more complete characterization.

4) More papers on the addition of Cu into the chemistry of HEAs needed to be reviewed to address the effect of the addition of elements that have positive mixing enthalpy. Reviewing the following papers could be helpful:

[a] Materials Science and Engineering: A 825, 141875

[b] Materials Letters 272, 127866

Response: The references have been reviewed and added to the manuscript. The authors are grateful to the reviewer for the contribution to improve the manuscript.

Reviewer 3 Report

  • Please check the whole text again there are some typing and format mistakes.
  • If the authors have EDS maps that would be useful to show those maps besides the BSE images of the different samples.
  • In Figure 3a and b would be useful to write the Miller indices above the peaks.

Author Response

  • Please check the whole text again there are some typing and format mistakes.

Response: The text has been revised carefully and some errors corrected

  • If the authors have EDS maps that would be useful to show those maps besides the BSE images of the different samples.

Response: There are not available EDS maps corresponding to the BSE images to perform a correlative analysis. The authors agree that EDS maps could provide an additional insight in to the obtained microstructures. The microanalysis performed in such BSE maps provided nevertheless enough information to characterize the microstructure obtained, and the approximate the composition in each of the features observed.

The paper contains instead EPMA measurements on the homogenized samples in order to understand the how the elements behave in solid solution. This has enable to observed the minor compositional variations that can be expected in the matrix.

  • In Figure 3a and b would be useful to write the Miller indices above the peaks.

Response: the Miller indices have been included.

Round 2

Reviewer 1 Report

The authors addressed all comments. Nice work. Acceptance is recommended.

Reviewer 2 Report

The revised manuscript could be considered for publication.